# Predicting disease risk areas through co-production of spatial models: The example of Kyasanur Forest Disease in India's forest landscapes

Bethan V. Purse[1]*, Narayanaswamy Darshan[2,3], Gudadappa S. Kasabi[2], France Gerard[1], Abhishek Samrat[4], Charles George[1], Abi T. Vanak[4,5,6], Meera Oommen[4,7], Mujeeb Rahman[4], Sarah J. Burthe[8], Juliette C. Young[8,9], Prashanth N. Srinivas[10], Stefanie M. Schäfer[1], Peter A. Henrys[11], Vijay K. Sandhya[2], M Mudassar Chanda[12], Manoj V. Murhekar[13], Subhash L. Hoti[3], Shivani K. Kiran[2]

1 UK Centre for Ecology and Hydrology, Wallingford, United Kingdom, 2 Department of Health and Family Welfare Services, Government of Karnataka, Shivamogga, India, 3 ICMR-National Institute for Traditional Medicine, Belgavi, India, 4 Ashoka Trust for Ecology and the Environment, Bengaluru, India, 5 DBT/ Wellcome Trust India Alliance Fellow, Hyderabad, India, 6 School of Life Sciences, University of KwaZulu-Natal, Durban, South Africa, 7 Dakshin Foundation, Bangalore, India, 8 UK Centre for Ecology & Hydrology, Edinburgh, United Kingdom, 9 Agroécologie, AgroSup Dijon, INRAE, Univ. Bourgogne Franche-Comté, Dijon, France, 10 Institute of Public Health, Bangalore, India, 11 UK Centre for Ecology and Hydrology, Lancaster Environment Centre, Lancaster, United Kingdom, 12 ICAR-National Institute of Veterinary Epidemiology and Disease Informatics, Bengaluru, India, 13 National Institute of Epidemiology (ICMR), Chennai, India

* beth@ceh.ac.uk

**Data Availability Statement:** The environmental predictor data can be accessed at https://doi.org/

## Abstract

Zoonotic diseases affect resource-poor tropical communities disproportionately, and are linked to human use and modification of ecosystems. Disentangling the socio-ecological mechanisms by which ecosystem change precipitates impacts of pathogens is critical for predicting disease risk and designing effective intervention strategies. Despite the global "One Health" initiative, predictive models for tropical zoonotic diseases often focus on narrow ranges of risk factors and are rarely scaled to intervention programs and ecosystem use. This study uses a participatory, co-production approach to address this disconnect between science, policy and implementation, by developing more informative disease models for a fatal tick-borne viral haemorrhagic disease, Kyasanur Forest Disease (KFD), that is spreading across degraded forest ecosystems in India. We integrated knowledge across disciplines to identify key risk factors and needs with actors and beneficiaries across the relevant policy sectors, to understand disease patterns and develop decision support tools. Human case locations (2014–2018) and spatial machine learning quantified the relative role of risk factors, including forest cover and loss, host densities and public health access, in driving landscape-scale disease patterns in a long-affected district (Shivamogga, Karnataka State). Models combining forest metrics, livestock densities and elevation accurately predicted spatial patterns in human KFD cases (2014–2018). Consistent with suggestions that KFD is an "ecotonal" disease, landscapes at higher risk for human KFD contained diverse forest-plantation mosaics with high coverage of moist evergreen forest and plantation, high

10.5061/dryad.tb2rbnzx5. The disease case data
cannot be shared publicly due to Reasonable
Security Practices and Procedures and Sensitive
Personal Data or Information Rules enforced by
Government of India. Researchers wishing to
access the human outbreak data used in this study
should contact the following officials at the
Department of Health and Family Welfare Services,
Government of Karnataka, Director (email:director-
hfws@karnataka.gov.in), Joint Director of
Communicable Diseases (CMD) (email:jdcmd-
hfws@karnataka.gov.in).

**Funding:** The MonkeyFeverRisk project that led to
these results is supported by the Global Challenges
Research Fund and funded by the MRC, AHRC,
BBSRC, ESRC and NERC [grant number MR/
P024335/1], awarded to BVP, SLH, MVM, MMC,
PNS, JY, SJB, MO, AV, FG, and GK. PAH and BVP
received additional support from NERC under the
SUNRISE project [grant number NE/R000131/1].
PNS received support from the Wellcome Trust/
DBT India Alliance Fellowship number IA/CPHI/16/
1/502648. The funders had no role in study design,
data collection and analysis, decision to publish, or
preparation of the manuscript.

**Competing interests:** The authors have declared
that no competing interests exist.

indigenous cattle density, and low coverage of dry deciduous forest. Models predicted new
hotspots of outbreaks in 2019, indicating their value for spatial targeting of intervention. Co-
production was vital for: gathering outbreak data that reflected locations of exposure in the
landscape; better understanding contextual socio-ecological risk factors; and tailoring the
spatial grain and outputs to the scale of forest use, and public health interventions. We
argue this inter-disciplinary approach to risk prediction is applicable across zoonotic dis-
eases in tropical settings.

## Author summary

Worldwide, impacts of zoonotic diseases, that cycle between animals and people, are con-
centrated in tropical communities and often linked to the way people use and change eco-
systems. Interventions for zoonotic diseases could be targeted better using risk maps
based on computer models that integrate social and ecological risk factors across degraded
ecosystems. However, such predictive models often perform poorly at local scales, incor-
porate narrow ranges of risk factors, and are disconnected from policy, managers and
interventions. Co-production brings together stakeholders and knowledge, across the
human health, animal health and environmental sectors, aligning with the OneHealth Ini-
tiative, to develop more informative predictive tools for zoonotic diseases. Through co-
production, we develop predictive models for a fatal tick-borne disease, Kyasanur Forest
Diseases (KFD) that is spreading across the degraded Western Ghats forest in India.
These models incorporating contextual risk factors identified by stakeholders, accurately
predicted patterns in human cases of KFD (2014–2018) in Shivamogga district, Karnataka
State, and identified new hotspots of infection during the subsequent 2019 outbreak.
Landscapes at highest risk encompassed diverse forest-plantation mosaics with high cov-
erage of moist evergreen forest and plantation, high indigenous cattle density, and low
coverage of dry deciduous forest. Co-production resulted in outbreak data that reflected
where exposure occurred in the landscape and outputs of value for targeting of interven-
tions, matched to the scale of forest use and public health interventions.

## Introduction

Zoonotic diseases disproportionately affect poor tropical communities[1–3], accounting for
around 26% of Disability-adjusted Life Years lost to infectious diseases in Lower Middle
Income Countries (LMICs). Communities affected by zoonoses often depend on surrounding
ecosystems for livelihoods and food security. In India, for example, around 300 million people
depend directly on degraded forest ecosystems for food, fuel, livestock fodder and other non-
timber forest products (NTFPs)[4]. A key cost of altering forest structure and accessing forest
goods and services is the increased exposure of humans and livestock to multi-host zoonotic
pathogens [5]. Forest habitats and their ecotones are a significant source of emerging and re-
emerging infections because they support complex ecological communities, including high
wildlife host and vector diversity [6–8]. Upsurges in incidence of several high burden zoonotic
diseases have been linked to deforestation or reforestation in LMICs (e.g. malaria[9], Leish-
maniases [10,11], Crimean-Congo Haemorrhagic Fever Virus) and to forest dependence. Liv-
ing in or near forests has been linked to unfair accumulation of geographic and social
disadvantages including political and economic marginalisation [12]. Forest communities are

rendered even more vulnerable by their remoteness from healthcare infrastructure [13]. Disentangling the ecological and social mechanisms by which changes in forest habitat can precipitate impacts of multi-host pathogens is critical for the design of effective intervention strategies.

Integrating land use patterns, ecosystem and social factors into interpretations of past disease patterns at a range of geographical scales [14] can indicate potential mechanisms and facilitate prediction of disease risk across new landscapes or for the same landscapes under future alternative environmental and development policies [15]. Conceptual frameworks, like the global "One Health" paradigm, recognise the interconnectedness of human health, wildlife and domestic animal health and the environment[16]. Despite this, models of zoonotic disease risk in tropical regions have often focussed on a single set of processes that drive variability in disease risk and effectiveness of interventions ([17] but see [14,15,18–20]). For example, published risk maps and spatial decision support tools for tick-borne zoonoses tend to focus on mapping environmental hazard (or presumed correlates of environmental hazard like tick abundance or presence[21]), but often do not integrate the social factors which drive patterns in exposure [20,22]. Tools and maps are rarely linked to intervention programs at a scale appropriate to sources of epidemics[17] and ecosystem use. Leach & Scoones [17] recommend instead that disciplines, data and models are not only integrated, but "triangulated" with deliberation around framing assumptions, policy narrative, politics and values. The process of co-production is ideally suited to the development of models to understand and predict zoonotic diseases. Co-production is based on the need to integrate different forms of knowledge into decision-making. It involves active engagement of stakeholders from different sectors and scales as knowledge holders and future model users through three key stages of framing the problem, knowledge integration and experimentation [23].

In our inter-disciplinary One Health Indo-UK partnership, the MonkeyFeverRisk project [24], co-production is used to improve understanding and develop decision support tools for a fatal tick-borne zoonotic disease, Kyasanur Forest Disease (KFD), that is spreading across degraded Western Ghats forest ecosystems in India.

Kyasanur Forest Disease Virus (KFDV; family Flaviviridae, genus Flavivirus) causes debilitating and fatal haemorrhagic disease (around 500 cases p.a., up to 10% mortality[25]) in forest communities. Key affected groups include small-holder farmers engaged in cultivation and grazing of cattle in forests [26], forest-dependent tribal communities who gather NTFP, day labourers in plantations and State forest department workers[27–29].

As well as affecting diverse human communities, the transmission cycle of KFDV is complex. KFDV cycles between different life stages of tick species from several genera (principally *Haemaphysalis* but also some *Ixodes* species) and amplifying vertebrate hosts including wild rodents and shrews, monkeys and some birds [25]. Humans contract KFDV when bitten by an infected tick, but are incidental hosts for the disease. Monkeys, principally the black-footed grey langur (*Semnopithecus hypoleucos*) and the bonnet macaque (*Macaca radiata*), are thought to act as amplifying hosts, by infecting large numbers of larval ticks with the virus [30]. Cattle do not amplify KFD since they do not develop viraemia of long duration [31], but may amplify tick populations through their importance as a blood meal host.

The emergence of KFD in humans has been widely linked to human modification of the forest ecosystem through deforestation [10,26,30]. The initial epidemics in Karnataka in the 1950s and those in the 1980s were preceded by population increases and extensive deforestation, to make way for plantations (such as Areca and cashew), paddy cultivation, housing and roads. This created mosaic tropical evergreen and deciduous forest, interspersed with cultivation (e.g. paddy), and interface scrub habitat between villages and forests that was conducive to both tick populations and cattle grazing [26,30,32]. These conditions are hypothesised to

have facilitated the emergence of KFDV into humans from a cryptic sylvatic enzootic cycle involving small mammals and monkeys [30]. However, it is still unknown how tick vectors and potential amplifying vertebrate hosts are linked to different habitats and to human exposure within agro-forest mosaics.

Human epidemics were restricted to focal areas of Karnataka State from 1957 to 2012, but since then human cases have been detected in four neighbouring states (Tamil Nadu, Kerala, Goa and Maharashtra)[33]. Human serological evidence indicates wide KFDV circulation in other states across India (Gujarat, West Bengal, the Andaman and Nicobar Islands) and on the border with China. Therefore, the landscape conditions favouring KFDV transmission are widespread [33], and the subset of these conditions that lead to human disease impacts, need to be delineated urgently. KFD impacts are managed currently through vaccination, awareness campaigns and promotion of tick protection measures in and around recently affected areas. However, constraints on availability and efficacy of the vaccine, and reluctance of local communities to be vaccinated and adopt personal protection measures can exacerbate epidemics [28,29,34]. Thus targeting of interventions towards the most vulnerable communities is critical.

This paper describes the co-production–with actors and beneficiaries across the public health, animal health and forestry sectors–of the landscape-level spatial models and understanding of risk factors for a case study zoonotic disease, Kyasanur Forest Disease. The model is developed for Shivamogga district in Karnataka, which has been affected by KFD since the 1950s, and reports a high proportion of India's human cases (e.g. 656 or 34% of 1929 cases reported between 2010 and 2019). Because of this, health managers of this district have long experience in disease surveillance and control. Using point locations for human cases recorded at sub-village level by health managers between 2014 and 2018, spatial Boosted Regression Tree models [35,36] are used to quantify the relative role of forest characteristics and loss, topography, host densities and public health factors in driving patterns in KFD at the landscape scale.

The co-production process involved framing potential key risk factors for KFD with cross-sectoral managers [37]. Spatial proxies of these risk factors were integrated into the model framework. The spatial grain of the model and its output was tailored to the scale at which people use forests (from household surveys) and the scale at which public health managers collect and report outbreak data. The models were then validated during the 2019 outbreak season with health managers, in terms of their predictive accuracy and utility for management.

Furthermore, through geographical thinning [38,39], the model framework accounted for the sparse, spatially clustered recording effort that often arises in public health surveillance datasets[40]. We discuss the extent to which disease patterns were predictable from landscape, topographical host and landscape metrics, whether human cases of KFD are associated with particular forest types, mosaic habitats or forest loss and how model predictions could improve targeting of interventions and surveillance.

## Methods

### Ethics statement

The protocols for this study were approved by the Institutional Ethics Committee of the Institute of Public Health (IPH IEC), Bangalore (Study ID, IEC-FR/04/2017) and received a Favourable Ethical Opinion from the Liverpool School of Tropical Medicine Research Ethics Committee (research protocol 17/062). All workshop participants were adults and provided informed consent via email through acceptance of the workshop invitation. The IPH IEC

approved the access and use of confidential patient data from DHFWS and all data were anonymised appropriately prior to analysis.

## Study area

Shivamogga covers an area of 8465 km$^2$ between 13.45$^o$ and 14.65˚ Latitude and 74.63$^o$ and 75.73˚ Longitude (Fig 1). The district is diverse in topography and vegetation, comprising Western Ghats mountains that are subject to high annual rainfall (900 to 8000 mm per annum) and drier inland plateau areas (range in elevation across Shivamogga is -70 m to 2674 m a.s.l., mean ± s.d. = 460 m ± 379 m a.s.l.), and a corresponding transition from evergreen and semi-evergreen forests, to moist deciduous forest and scrub. The forests of the Western Ghats have been degraded and fragmented throughout the 19[th] and 20[th] centuries, due to timber extraction, industrial development (roads, railways, dams and mines) and increases in agriculture and plantations [41,42]. The area of forest vegetation in Shivamogga has declined from an estimated 43.8% of the district in 1973 to 22.3% of the district in 2012, producing patch and edge forest [43].

## Human case data

Human cases of Kyasanur Forest Disease occur seasonally between December and May when the abundance of infected nymphal ticks in the forest is at a peak. Designated laboratories for processing human samples of KFD from Shivamogga District include the Virus Diagnostic Laboratory (VDL), Shivamogga, the ICMR-National Institute of Virology, Pune and Manipal Centre for Virus Research. Human cases, arising from samples testing positive for Kyasanur Forest Disease by RT-PCR or IgM ELISA in either laboratory in the five years between the December 2013 / May 2014 and December 2017 / May 2018 outbreak seasons, were compiled by co-author, Dr S. K. Kiran, who served as Taluka Medical Officer for Tirthahalli from 2010–2019. Cases were assigned to locations retrospectively using Google Earth, following personal visits to households of affected patients conducted by Dr Kiran, or Medical Officers in other talukas, during the outbreak seasons. These locations were marked on Google Earth to retrieve the geographical coordinates in Latitude and Longitude to an estimated spatial precision of around 300 m. Cases of febrile illness may be reported to Primary Health Centres and from home addresses that are very distant from where infection is acquired. For example, migrant agricultural labourers work in plantations and pilgrims visiting temples in forest areas that can be 10s to 100s of kilometres from their homes [29]. Based on case-tracing that Taluka Medical Officers had performed during the outbreak seasons, such cases (< 10) could be excluded from the analysis. In total, 329 cases from 117 different household or village locations were compiled over the five transmission seasons (144 cases from 2013/2014, 32 from 2014/2015, 32 from 2015/2016, 89 from 2016/2017, 32 from 2017/2018). The presence and number of cases were summarised at a 1 km x 1 km and 2 km x 2 km grid resolution across Shivamogga, resulting in 65 (of a total of 7732) land cells and 53 (of a total of 1926) land cells positive for KFD respectively at these study grains. These two study grains were chosen to reflect the range of distances from households at which forest users may acquire infection from forest habitats during their livelihood activities. Interviews with members of similar forest communities in Wayanad, Kerala revealed that forest users move between 1 and 4 km through forest habitats from their homes during the main risk KFD period(January to March, see S1 File).

During the 2018/2019 transmission season in Shivamogga District, households with cases throughout Shivamogga District were geo-located as a passive independent validation dataset. Between 21 November 2018 and 16 June 2019, 344 human cases of KFD were reported across 104 villages, causing 20 deaths. The majority of these cases (212 cases or 61%) and 17 deaths

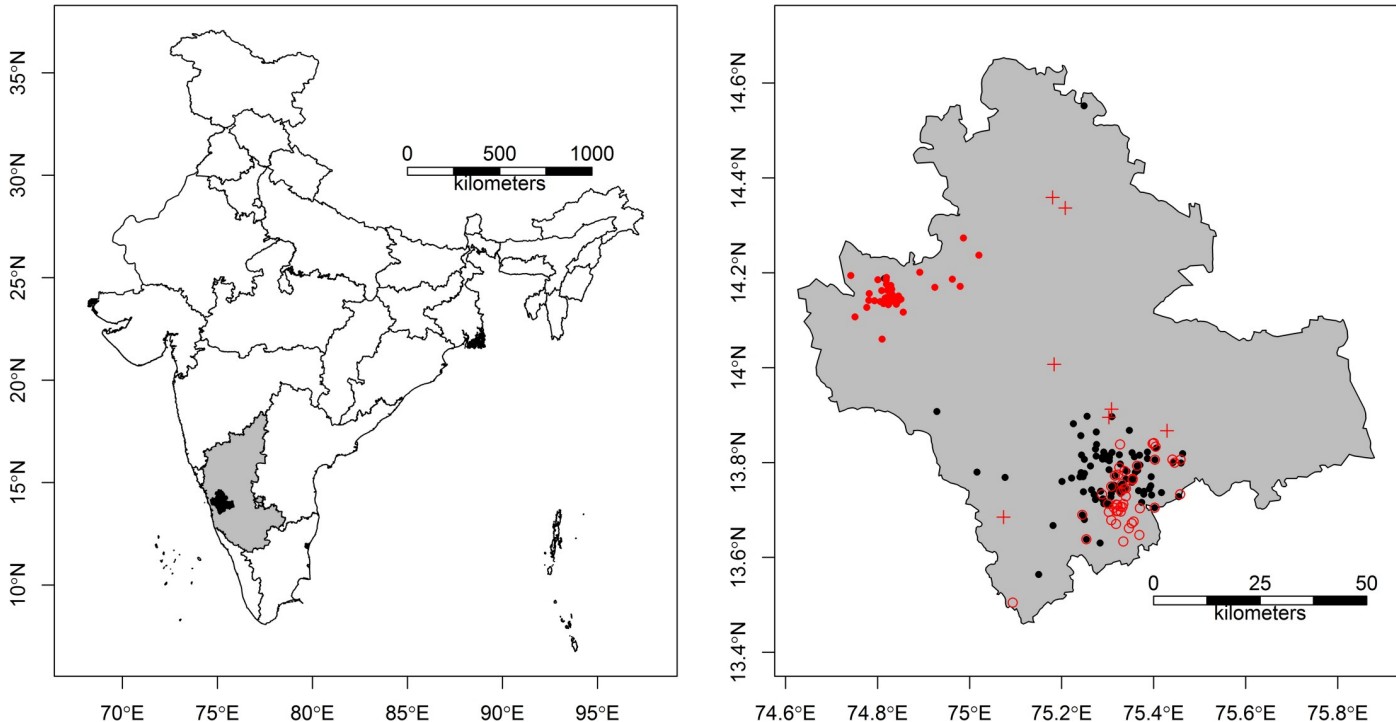

**Fig 1.** (a) Map of India depicting the location of Shivamogga district (black shading) within Karnataka State (grey shading). (b) Map of Shivamogga district showing locations of households with human cases in black (2014–2018 seasons) and red (2018–2019 season). Amongst the 2018–2019 cases, closed red circles are affected households in Sagara taluka and open red circles are affected households in Tirthahalli taluka, whilst crosses are affected households from other talukas. The administrative boundary dataset is from HindustanTimesLabs (https://github.com/HindustanTimesLabs/shapefiles/), reproduced under the MIT License. Note that Bhadravathi taluk, in the southeast corner of Shivamogga district, is omitted from the study.

were concentrated in a new geographical focus in Sagara taluka in northern Shivamogga (Fig 1B, closed red circles). This area had reported no human cases and only one or two human or tick positives in the 2014/2015 and 2017/2018 season during the prior decade. Geo-location of affected households was achieved directly in the field using the Smartphone Android App "AndLocation" or by health workers who shared their live location on WhatsApp with project staff for transfer to Google Earth and retrieval of coordinates. These 2018/2019 season cases spanned 84 of the 1 km grid cells and 68 of the 2 km grid cells. They provided an ideal test of whether the risk maps developed before the transmission season (using human cases data from the prior five transmission seasons) are capable of predicting new outbreaks, including new geographical foci like the one in Sagara taluka.

## Framing key socio-ecological risk factors for KFD with stakeholders

The participatory MonkeyFeverRisk Framing workshop was held on 16th August 2018 in Bengaluru, Karnataka, India. It involved over 20 experts from different KFD-affected districts and states level of Karnataka, Maharashtra and Kerala, including officials from the public and animal health, agriculture, forestry and social welfare sectors [37]. Participants of the workshop were selected based on a stakeholder mapping exercise. This identified key actors from different sectors and working at different scales, likely to play major roles in the understanding and management of KFD. The two aims of the workshop were to (i) identify the key risk factors for KFD as prioritized by stakeholders and (ii) identify key policies that affect KFD transmission and management using participatory approaches, as outlined in S2 File. The key risk factors

for KFD that received four or more votes during the ranking exercise among stakeholders are shown in Table 1, together with their links to particular spatial environmental predictors in the analysis (right hand column). The full suite of gridded spatial environmental predictors that was generated within topographical, landscape, host and public health categories is shown in Table 2, including additional risk factors drawn from scientific literature on KFD epidemiology and ecology. The sources and processing methods for selected spatial environmental predictors, including measures to account for collinearity between predictors, are detailed in S3 File.

Forest loss and degradation was the highest ranked environmental risk factor for KFD by stakeholders, consistent with scientific literature linking forest loss and creation of mosaic forest-paddy-scrub village habitat to human emergence events [26,30,32]. Stakeholders considered that abrupt shifts in land use between forest and village areas made communities more vulnerable to KFD. Metrics of forest change considered in the analysis were area of forest loss and gain since 2000 per grid cell derived from Hansen et al. [44]. Area of forest gain was highly collinear with area of forest loss (Pearson's correlation coefficient, r = 0.968), and much less prevalent (making up 0.16% versus 1.2% of 30m land pixels), so was excluded from the analysis. To quantify mosaic habitat, the amount and diversity of different forest types, of agricultural or fallow land and plantation and overall land use diversity were extracted from the MonkeyFeverRisk Land Use Land Cover map (LULC map), derived from Landsat Thematic Mapper imagery (2016–2017) as detailed in S4 File. Edge metrics for forest types were also calculated as a measure of the amount of interface habitat between forest, agriculture and villages but were highly collinear with amount of individual forest types and so were not included in the final analysis (S3 File).

Stakeholders also ranked human use of forests and living in/around forests as key risk factors. They identified policies (or poor policy implementation) linked to grazing and encroachment in and around forest areas as increasing the risk of KFD. This is consistent with case-control studies that have linked grazing cattle inside forests, handling cattle and gathering of dry leaves for animal bedding to higher exposure to KFD [29] and the hypothesised role of cattle in amplifying tick populations. Available spatial proxies for extent of forest use for grazing were the densities of indigenous cattle (since smallholders in Shivamogga keep indigenous breeds) and buffalos per grid cell.

Again consistent with literature [28,34], diverse public health factors were ranked by stakeholders as key risk factors for KFD (Table 1) such as lack of awareness of KFD and preventative measures, low acceptance and coverage of vaccination, poor diagnostics and surveillance including under-reporting of human cases or monkey deaths. Available spatial proxies of access to health services and education and surveillance effort, developed from Indian Government census data (S3 File), were the proximity to a primary health centre, overall human population size and the number of medics available per head of population at village level (S3 File). Some risk factors like poor data management, low vaccine uptake and under-reporting of monkey deaths by the Forest Department, will be less well linked to such spatial proxies.

Consistent with literature linking tick demography and host-seeking behaviour to micro-climatic factors [21,45,46], stakeholders also mentioned suitable micro-climates for tick populations among key risk factors for KFD (Table 1). However, available gridded weather station data is too coarse in resolution to define spatial variation in climate at village to district levels. Topographical factors, namely slope and elevation, and associated vegetation types from the LULC map were considered to be better proxies of micro-climatic conditions at village and district scale. Small water bodies of around 10m$^2$ were hypothesised by disease managers to be key locations in the landscape where monkeys carrying KFD, nymphal ticks and grazing animals might co-occur. Conversely, large water-bodies could constitute barriers to dispersal of

**Table 1. Ranked list of key risk factors for Kyasanur Forest Disease produced by cross-sectoral stakeholders during the MonkeyFeverRisk Problem Framing workshop.**

| Rank | Risk factors | Number of votes | Spatial proxies in models that link to each risk factor |
|---|---|---|---|
| 1 | Lack of awareness about KFD | 10 | Proximity to health centre, number of medics per head of population, human population density |
| 2 | Under or late reporting of monkey deaths | 9 | None |
| 2 | Deforestation and/or forest degradation | 9 | Forest loss, human population density, cover of agriculture and plantations, forest and land use diversity |
| 2 | Lack of awareness of preventative measures (tick repellants, vaccination) | 9 | Proximity to health centre and number of medics per head of population |
| 3 | Lack of understanding of alternative hosts | 8 | None, though alternative hosts linked to forest types |
| 4 | Human use of forests | 7 | Cover and diversity of forest types |
| 4 | Low vaccination coverage | 7 | Both factors expected to be linked to Proximity to health centre, number of medics per head of population, human population density |
| 4 | Poor diagnostics and surveillance | 7 | |
| 4 | Lack of OneHealth policy | 7 | None |
| 5 | Poor data management | 6 | None |
| 5 | Poor understanding of tick ecology | 6 | None |
| 6 | Side effects and concerns about vaccines | 5 | None |
| 7 | Living in or around forests | 4 | Cover and diversity of forest types |
| 7 | Favorable environment for ticks | 4 | Cover and diversity of forest types, micro-climate availability linked to topography |
| 7 | Poor tick identification | 4 | None |

**Table 2. Potential environmental predictors included in models of Kyasanur Forest Disease distribution†.**

| Category of predictor | Predictor name(abbreviation) | Description / units | Mean and s.d. across the region (1 km) | Range across the region (1 km) |
|---|---|---|---|---|
| TOPOGRAPHY | Elevation (elev) | mean elevation (m. a. s. l.) | 559 ± 190 | 2–1224 |
| | Slope (slope) | mean slope (degree) | 4.5 ± 3.7 | 0–32.1 |
| LANDSCAPE CHANGE | Area of forest loss | proportional area of cell classified as forest lost during 2000–2014 | 1.1 ± 2.6 | 0–43 |
| LANDSCAPE | Forest type diversity | diversity of forest types(Shannon-Weaver Index accounting for % area per cell per forest type) | 0.77 ± 0.32 | 0–1.37 |
| | Area of dry deciduous | proportional area of dry deciduous forest per cell | 13 ± 16 | 0–100 |
| | Area of wet deciduous | proportional area of moist deciduous forest per cell | 20 ± 22 | 0–100 |
| | Area of wet evergreen | proportional area of wet evergreen forest per cell | 17 ± 24 | 0–100 |
| | Area of plantation | proportional area of plantation per cell | 15 ± 13 | 0–90 |
| | Area of water bodies | proportional area of waterbody per cell | 4 ± 14 | 0–100 |
| | Area of agricultural or fallow land | proportional area agricultural or fallow land | 25 ± 24 | 0–96 |
| HOSTS | Buffalo density | buffalo density in mean head per km per cell | 17.5 ± 17.0 | 0.0–245.8 |
| | Cattle density | indigenous cattle density in mean head per km per cell | 60.0 ± 40.5 | 0.1–904.0 |
| | Human population density | human population size in mean head per km per cell | 204.7 ± 528.8 | 0–7087.8 |
| PUBLIC HEALTH | Proximity to primary health centre (PHC_proximity) | proximity to primary health centrewhere 1 = Primary Health Centre (PHC) within village, 2 = PHC > 5 km, 3 = PHC 5–10 km, 4 = PHC > 10 km from village | 3* | 1–4 |
| | No. of medics per head of population (Nmedics) | no. of medics per head of population | 0.0006 ± 0.0018 | 0–0.04 |

†See S3 File for the sources and processing methods for these environmental predictors.
*modal value.

wildlife hosts for tick vectors and KFD, and in turn, to epidemic spread [47,48]. The LULC map is based on 30 m resolution Landsat data, thus only detects water-bodies of larger than 30m. Thus, the coverage of such larger water-bodies per grid cell was included a priori amongst predictors, with the expectation that high coverage of such water-bodies would be unfavourable for KFD.

## Modelling the distribution of Kyasanur Forest Disease with Boosted Regression Trees

A boosted regression tree (BRT) modelling [35,36] framework was used to determine the sensitivity of patterns in human KFD cases to land use, topographical and host variability, and to generate maps of potential distribution across Shivamogga. BRTs combine regression trees, which build a set of decision rules on the predictor variables by portioning the data into successively smaller groups with binary splits [35,36], and boosting, which selects the tree that minimises the loss of function, to best capture the variables that define the distribution of the input data. BRTs have been shown to have high performance amongst methods used to predict species distributions [49], probably due to their ability to fit complex, non-linear responses to environmental covariates and their robustness to outliers. However, they can be prone to over-fitting data and therefore a number of stringent cross-validation checks were used to avoid this (see below).

Due to the high degree of spatial clustering in the case presence data (Fig 2), it was clear that this presence data should be thinned prior to analysis to avoid inflation of model accuracy and pseudo-replication of particular environmental conditions in the model [38,39]. Presence data can be thinned in geographical or environmental space [38]. The lack of quantitative information on the environmental drivers of KFD makes it difficult to select key environmental axes by which to thin the data. Thus thinning in geographical space was conducted. A rule of thumb is to take the peak distance between pairs of presence records (this is 10 km for KFD in S1 Fig) and thin the points so that they are no closer than half this distance (~ 5 km for KFD), ensuring that there is only 1 presence in each 5 km grid cell. Thus, at each resolution (1 km grid cell and 2 km grid cell), we randomly selected one of the presence records per 5 km grid cell, for each of the 30 x 5 km grid cells in which presence records were found. The whole process was repeated 100 times to generate 100 presence datasets at each resolution.

Absence data: Since Boosted Regression Trees require both presence and absence data, the presences were matched with an equal number of selected absence grid cells as recommended by Barbet-Massin et al.[50] for BRTs. When selecting absence data, it is important to try to mimic the recording process that gives rise to the presence data. For example, human cases of KFD tend to be reported from rural land areas and communities. Therefore very large water bodies and built-up areas like towns and cities were excluded from the absence selection process by including only 1 km or 2 km grid cells encompassed by the census village boundaries in the selection area. Cells containing presences were also removed from the selection area at each resolution. We then randomly selected 30 absence cells at each resolution, each of which occurred in a different 5 km grid cell, and this process was repeated 100 times to generate 100 absence datasets. Each presence dataset was combined with one of the absence datasets to generate 100 presence-absence datasets at each resolution.

Due to the disparity between local land use maps and global forest loss data (S3 File), at each resolution, a set of 100 BRT sub-models was fitted to all environmental predictors and another set fitted to all predictors except area of forest loss. Models were fitted using the gbm. step function of the dismo package in R [51]. Model settings were as follows: learning rate = 0.001; tree.complexity = 4; bag rate = 0.6; to allow two way interactions between

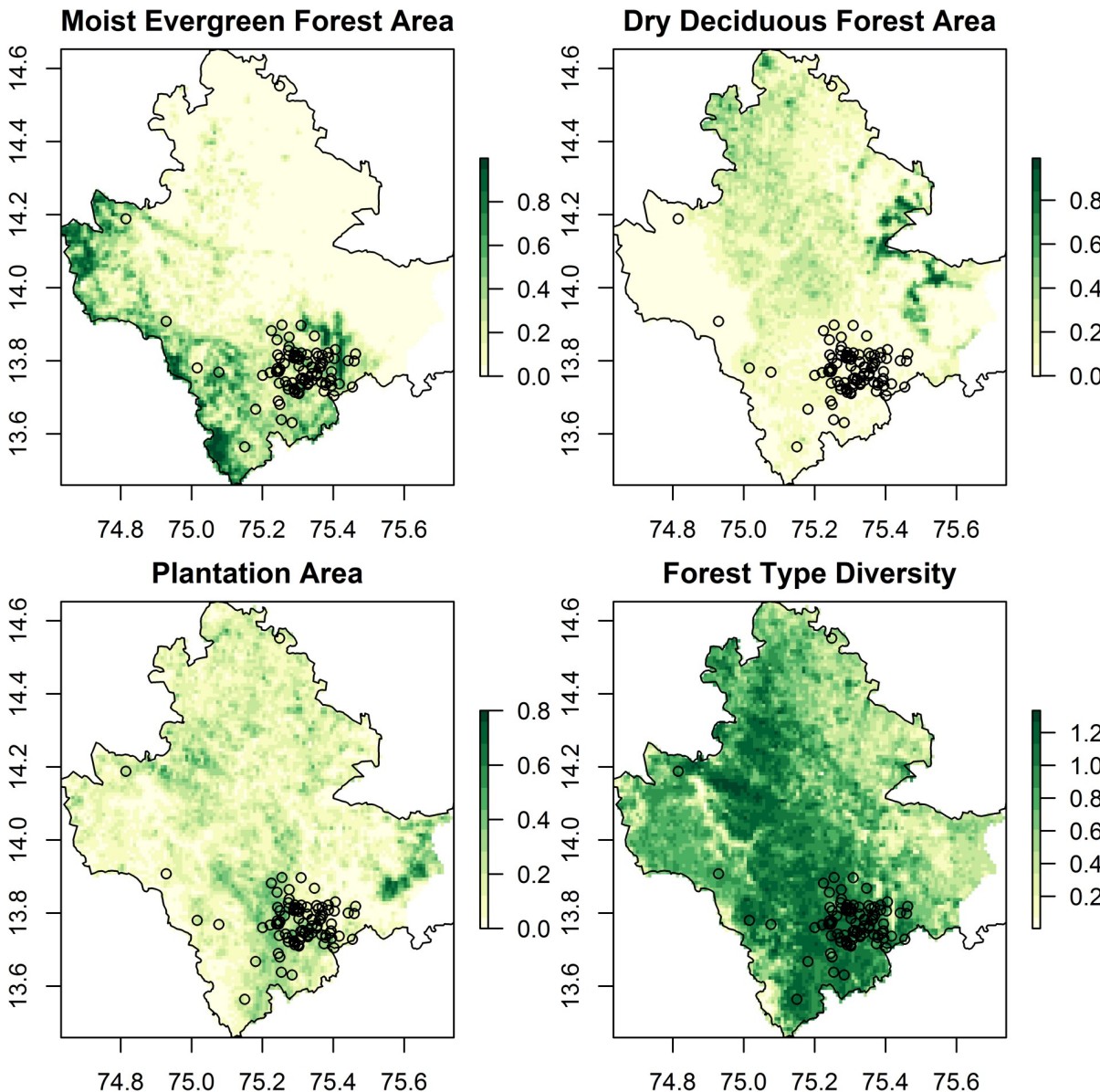

**Fig 2. Key landscape predictors of presence of Kyasanur Forest Disease (1 km resolution) overlaid with point locations of human cases from 2014 to 2018 (black dots).).** These are proportional areas of moist evergreen forest, dry deciduous forest, and plantation per grid cell and forest type diversity. These metrics were derived from analysis of the MonkeyFeverRisk LULC map (see S3 and S4 Files). The administrative boundary dataset is from HindudstanTimesLabs (https://github.com/HindustanTimesLabs/shapefiles/), reproduced under the MIT License. Human case data are from the Department of Health and Family Welfare Services, Government of Karnataka. White areas in Bhadravathi taluka in southeast corner indicate no data.

environmental predictors and to ensure that optimum number of trees always exceeded 1000. The gbm.step function automatically identifies the optimum number of trees for a BRT model using ten-fold cross-validation, selecting the number of trees that minimises hold-out deviance (cross-validation deviance) across folds. In addition to the cross-validation deviance, gbm.step reports several metrics of model performance in cross-validation across folds including; (i) the Area Under the Receiver Operator Curve Statistic (AUC) on hold-out dataset [52], or cross-validation AUC, where an AUC value of 0.5 indicates no discriminative ability between

presence and absence, and a value of 1 indicates perfect discrimination; (ii) cross-validation coefficient, which is the Pearson's correlation coefficient between the predicted probability of presence and the true presence/background for the hold-out dataset. We report the mean, median and standard deviation of these metrics across the 100 BRT sub-models in each model set. This provides a full picture of the average performance and consistency across each model set. The true prevalence of the disease across the study region is unknown, and the models are parameterised with ad hoc presence data combined with selected pseudo-absences rather than true absences. Therefore, the models predict the relative rather than absolute probability of presence between cells. This model fitting process was repeated at each resolution for the two sets of environmental predictors as above, giving rise to four different model runs- 1 km with forest loss, 1 km without forest loss, 2 km with forest loss, and 2 km without forest loss.

Relative contribution statistics of predictor variables are reported only for the BRT model with the optimum number of trees (not for the folds). Relative importance is defined as the number of times a variable is selected for splitting, weighted by the squared improvement to the model as a result of each split and averaged over all trees [52]. These contributions are scaled to sum to 100, with a higher number indicating a greater effect on the response. Again, we report the average of these values across the 100 BRT sub-models.

The direction of the association between human cases of KFD and particular predictor variables was evaluated from the response curves produced from the BRT model. The response curves for a predictor were averaged across the 100 BRT sub-models by calculating mean and standard deviation of the marginal predicted probabilities within ~40 bins of the predictor values.

The extent to which geographical thinning was successful in removing spatial autocorrelation caused by clustering of presence records was examined by plotting correlograms of the residuals of each fitted sub-model using the correlog function in the ncf package in R [53]. The Moran's I values and significance values were then averaged across sub-models within different distance bins (from 0 to 80 km in increments of 5 km), to look for evidence of systematic positive spatial autocorrelation.

To predict the distribution of KFD across Shivamogga in unsampled areas, each BRT sub-model was applied to the prediction layers for a given resolution and environmental set using the predict.gbm function of the gbm package in R [54]. The predicted relative probability of presence was averaged across sub-models to produce an ensemble mean ± standard deviation of the relative probability of presence per grid cell. For each BRT model, the threshold relative probability of presence that maximises discrimination between presence/background for the hold-out dataset was calculated by gbm.step and averaged across folds (cross-validation threshold). Occurrence patterns were summarised across sub-models by converting the predicted distributions to binary presence-absence maps per sub-model using the mean cross-validation threshold relative probability of presence for each sub-model, and counting the number of times a cell was predicted to be present across the 100 sub-models. The predicted extent of occurrence in terms of number of study grid squares or pixels was calculated for each predictor resolution and environmental set. The geographical extent of predictions was limited to the region for which the environmental predictors were available and, out of necessity, omitted very large water-bodies and areas classified as towns or cities in the census.

Independent validation of models was achieved using the presence points from the 2018/2019 outbreak, by calculating the predicted-to-expected (P/E) ratio for all model sets using the ecospat.boyce function of the ecospat package in R. Here a graph of predicted versus expected for a good model should show a monotonically increasing curve [55], and the correlation between P/E and habitat suitability should be positive if model predictions are consistent with the distribution of presences in the independent validation dataset [56].

Not only does this independent data test the validity of the models, but also the assumption of stationarity that cannot be tested through cross validation. That is to say, the assumption that any relationships derived from the observed data hold true and are consistent for areas/periods beyond the range of observed data. This assumption is crucial if derived models are to be used for any early warning system.

The validation was first conducted using all the 2018 to 2019 season case locations (n = 84 cells at a 1 km resolution, n = 68 cells at a 2 km resolution). Then, to test whether the models could predict new geographic foci of human cases, we conducted the validation separately and compared results for: (i) locations from the newly affected sub-district of Sagara (61% of 2018 to 2019 season case locations, spanning 37 cells at a 1 km resolution and 25 cells at a 2 km resolution); and (ii) locations from the sub-district of Tirthahalli (36% of 2018 to 2019 season case locations, spanning 40 cells at a 1 km resolution and 36 cells at a 2 km resolution), which recorded 95% of the past cases used to parameterise the model.

## Results

### Environmental predictors of the recent past distribution of Kyasanur Forest Disease

Models combining topography, landscape metrics, hosts, and public health predictors predicted recent patterns in KFD with a high degree of accuracy. Values of AUC in cross-validation ranged from 0.85 to 0.90 (Tables 3 & 4). Models had a similarly high accuracy whether area of forest loss was included or not.

At a 1 km resolution, predictors which were consistently most important in predicting presence of KFD are the area of moist evergreen forest, the diversity of forest types followed by the density of indigenous cattle and the area of dry deciduous forest (Table 5, left hand columns). Elevation and the area of plantation were also often ranked highly in importance amongst predictors. The geographical variability in these key predictors in relation to the 2014 to 2018 distribution of cases is depicted in Figs 2 and 3.

The response plots in Fig 4 indicate that KFD outbreaks are more likely to occur when the proportional area of moist evergreen forest is intermediate or high (>15% of the area), when the proportion of dry deciduous forest is low (<10% of the area), when the amount of forest loss pixels is higher (> 10% of the area), when indigenous cattle densities are higher, when forest diversity is higher (>1.0) and the proportional area of plantation is higher. Figs 2 and 3 illustrate how the cases between 2014 and 2018 were indeed concentrated in 1 km squares with a diverse mosaic of forest types, namely moist evergreen and plantation that are likely to have undergone rapid change. Once area of forest loss was added to the model, it ranked second in importance to the area of moist evergreen forest but the subsequent order of importance of the other landscape predictors listed above remained the same as for models without forest loss (Table 5, right hand columns). The addition of area of forest loss to the models at 1 km does not increase the accuracy in cross-validation over models without area of forest loss.

The results at a 2 km resolution were similar in that the area of moist evergreen forest again far outranked the other predictors with a relative importance of around 20% and modal ranked importance of 1 (Table 6). The area of dry deciduous forest and area of plantation were next in importance followed by forest diversity and elevation. Once area of forest loss was added to models at a 2 km resolution, it again ranked second in importance to the area of moist evergreen forest but the subsequent order of importance of the other landscape predictors listed above remained the same as for models without forest loss (Table 6, right hand columns). Again, the addition of forest loss to the models at 2 km did not increase the accuracy in cross-validation over models without forest loss.

**Table 3. Accuracy metrics for BRT models of Kyasanur Forest Disease distribution—1 km resolution.**

| Metric | 1 km without forest loss metrics | | | 1 km with forest loss metrics | | |
|---|---|---|---|---|---|---|
| | median | mean | s.d. | median | mean | s.d. |
| Number of trees | 1775 | 1917 | 884 | 1775 | 1891 | 945 |
| Total deviance | 1.386 | 1.386 | 0.000 | 1.386 | 1.386 | 0.000 |
| Residual deviance | 0.386 | 0.406 | 0.209 | 0.358 | 0.399 | 0.215 |
| Cross validation deviance (mean) | 0.937 | 0.946 | 0.184 | 0.944 | 0.946 | 0.178 |
| Cross validation deviance (standard error) | 0.145 | 0.144 | 0.044 | 0.136 | 0.140 | 0.036 |
| Training set correlation | 0.930 | 0.919 | 0.057 | 0.940 | 0.920 | 0.060 |
| **Cross-validation correlation** | **0.673** | **0.650** | **0.119** | **0.686** | **0.654** | **0.116** |
| Cross validation correlation (standard error) | 0.083 | 0.087 | 0.030 | 0.082 | 0.084 | 0.027 |
| Training set AUC | 0.997 | 0.989 | 0.015 | 0.997 | 0.989 | 0.016 |
| **Cross validation AUC** | **0.867** | **0.851** | **0.070** | **0.867** | **0.857** | **0.069** |
| Cross validation AUC (standard error) | 0.051 | 0.052 | 0.017 | 0.050 | 0.050 | 0.016 |
| Cross validation threshold | 0.533 | 0.533 | 0.027 | 0.537 | 0.537 | 0.029 |

Response plots (see S5 File) indicate that KFD outbreaks are more likely to occur when the proportional area of moist evergreen forest is intermediate or high (>20% of the area), the proportion of dry deciduous forest is low (<10% of the area), the amount of forest loss is higher (> 10% of the area), elevation exceeds 650–675 m.a.s.l., forest diversity is higher (>1.0) and proportional area of plantation is higher.

There was considerable variation in the rank importance of variables across model runs for all model sets (see rank columns in Tables 5 and 6) which is to be expected given the low sample size that results from the geographical thinning. In all model sets, the public health predictors (proximity to health centres, number of medics per head of population) and human population size had consistently low importance.

In tests for spatial autocorrelation in model residuals, significant but low magnitude (Moran's I < = 0.2) was found at distances between 1 and 15 km but this was not consistent across

**Table 4. Accuracy metrics for BRT models of Kyasanur Forest Disease distribution—2 km resolution.**

| Metric | 2 km without forest loss metrics | | | 2 km with forest loss metrics | | |
|---|---|---|---|---|---|---|
| | median | mean | s.d. | median | mean | s.d. |
| Number of trees | 950 | 1095 | 545 | 950 | 1064 | 492 |
| Total deviance | 1.386 | 1.386 | 0.000 | 1.386 | 1.386 | 0.000 |
| Residual deviance | 0.299 | 0.323 | 0.161 | 0.290 | 0.313 | 0.155 |
| Cross validation deviance (mean) | 0.837 | 0.825 | 0.170 | 0.848 | 0.819 | 0.164 |
| Cross validation deviance (standard error) | 0.140 | 0.146 | 0.038 | 0.144 | 0.148 | 0.038 |
| Training set correlation | 0.938 | 0.930 | 0.043 | 0.945 | 0.934 | 0.040 |
| **Cross-validation correlation** | **0.731** | **0.723** | **0.090** | **0.727** | **0.721** | **0.088** |
| Cross validation correlation (standard error) | 0.072 | 0.074 | 0.025 | 0.075 | 0.078 | 0.029 |
| Training set AUC | 0.997 | 0.993 | 0.009 | 0.998 | 0.994 | 0.008 |
| **Cross validation AUC** | **0.900** | **0.900** | **0.048** | **0.900** | **0.897** | **0.051** |
| Cross validation AUC (standard error) | 0.041 | 0.042 | 0.015 | 0.044 | 0.045 | 0.017 |
| Cross validation threshold | 0.536 | 0.540 | 0.035 | 0.538 | 0.537 | 0.029 |

**Table 5. Relative importance (RI) and modal rank (rank) of predictors in BRT models at a 1 km resolution (median, mean and s.d. of values across 100 model runs).**

| Predictor | 1 km resolution without forest loss metrics | | | | 1 km resolution with forest loss metrics | | | |
|---|---|---|---|---|---|---|---|---|
| | median RI | mean RI | s.d. RI | rank | median RI | mean RI | s.d. RI | rank |
| Area of moist evergreen forest | 13.9 | 15.9 | 9.1 | 1 | 12.0 | 13.6 | 9.1 | 1 |
| Area of forest loss | - | - | - | - | 10.6 | 12.9 | 8.6 | 1 |
| Indigenous cattle density | 11.5 | 11.9 | 6.7 | 2 | 10.0 | 10.5 | 6.3 | 2 |
| Forest diversity | 10.7 | 13.4 | 8.8 | 1 | 9.2 | 11.4 | 8.7 | 1 |
| Area of dry deciduous forest | 10.2 | 12.1 | 8.1 | 3 | 8.8 | 10.7 | 7.6 | 5 |
| Elevation | 9.1 | 9.7 | 5.5 | 4 | 7.7 | 8.8 | 5.2 | 4 |
| Area of plantation | 8.6 | 11.0 | 7.4 | 4 | 7.3 | 8.8 | 6.3 | 4 |
| Slope | 6.5 | 8.2 | 5.8 | 8 | 5.7 | 7.2 | 5.3 | 9 |
| Buffalo density | 3.0 | 3.6 | 2.6 | 9 | 2.7 | 3.4 | 2.6 | 10 |
| Area of agricultural or fallow land | 2.9 | 3.7 | 2.6 | 9 | 2.6 | 3.4 | 2.4 | 10 |
| Area of moist deciduous forest | 2.9 | 3.70 | 3.1 | 1 | 2.4 | 3.2 | 2.7 | 11 |
| Human population density | 2.3 | 2.79 | 1.9 | 11 | 2.1 | 2.6 | 2.0 | 12 |
| Area of waterbodies | 1.6 | 2.47 | 2.4 | 12 | 1.6 | 2.2 | 2.4 | 13 |
| Proximity to primary health centre | 0.5 | 1.03 | 1.1 | 13 | 0.6 | 1.0 | 1.1 | 14 |
| Number of medics per head of population | 0.0 | 0.5 | 1.2 | 14 | 0.0 | 0.4 | 1.1 | 15 |

model runs. Inference from the BRT models is more robust to small spatial correlation due to the flexibility of the modelling approach, the robustness to outliers and the use of importance metrics rather than significance testing. Therefore we expected this remaining small spatial autocorrelation to have a negligible impact on the above inferred role of environmental predictors and estimates of model accuracy (S6 File).

## Predicted distribution of Kyasanur forest disease

Geographical patterns in areas predicted to have a high probability of KFD presence are similar between 1 km and 2 km grid cell resolutions (S7 File). There was also a high degree of correlation between mean predicted probability of presence layers between models with and without forest loss (Pearson's r = 0.974, p < 0.005 at 1 km).

Since the area predicted to be suitable for KFD varies between model runs, it is useful to look at the orange and red cells in the bottom panel of the prediction figure, which are cells in which KFD is predicted to be present in more than half of model runs. Suitable habitat for KFD is predicted to occur across a wide area in the south and south east of the district, extending between 5 km and 10 km around current presence locations (see Fig 5 for models at 1 km without forest loss, S7 File for all other model sets). Pockets of suitable habitat for KFD are predicted to occur also to the south and north of the large water-bodies (white cells) in the east of the district and in sporadic isolated locations in the north. The absence of KFD in agricultural areas along the northeastern band of Shivamogga (blue areas in all panels) and the southwestern coast is widely predicted across the models.

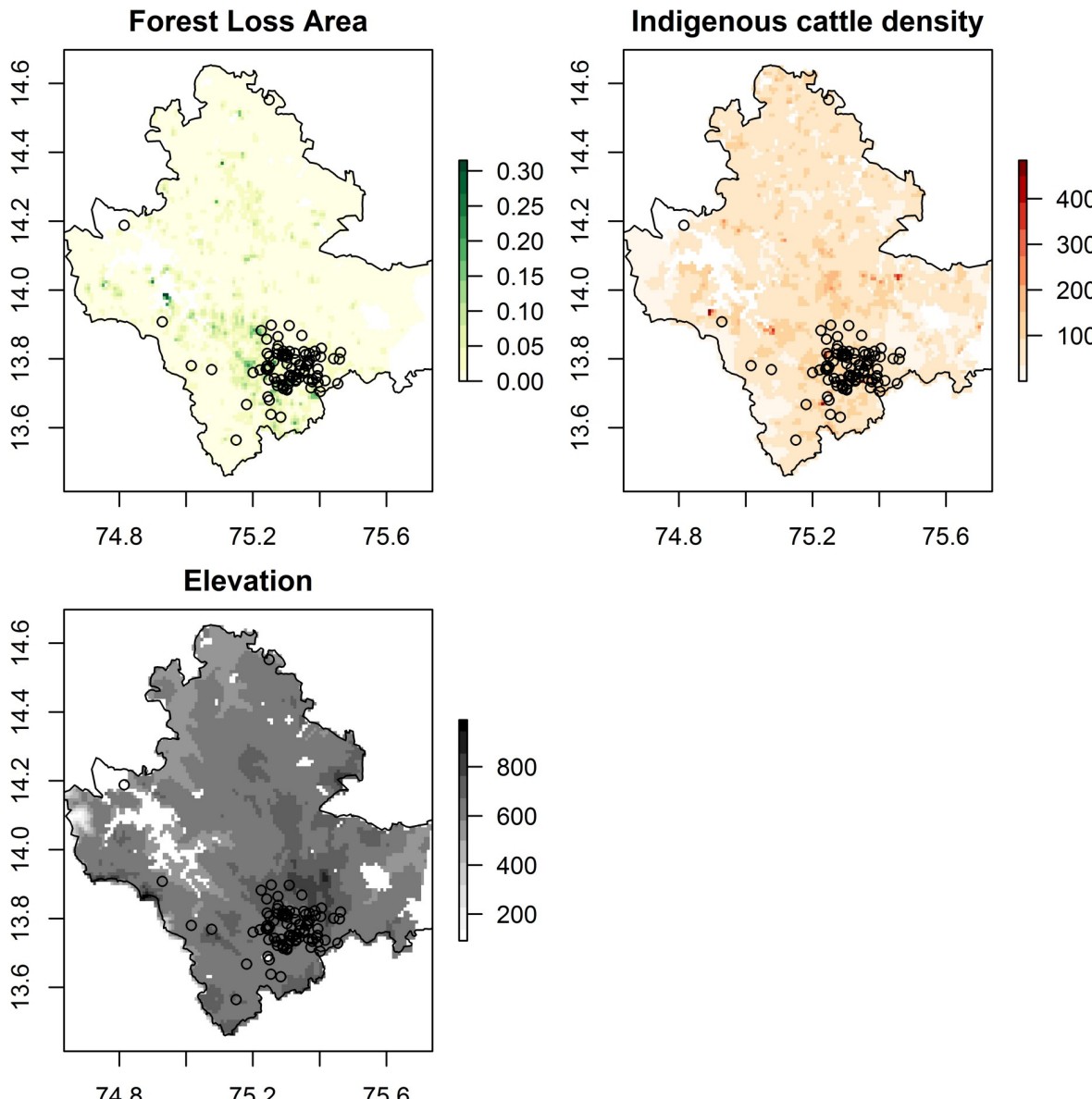

**Fig 3. Other key predictors of presence of Kyasanur Forest Disease (1 km resolution) overlaid with point locations of human cases from 2014 to 2018 (black dots).** These are area of forest loss, densities of indigenous cattle and elevation derived from Hansen et al. [44], from the Department of Animal Husbandry, Dairying and Fisheries, Government of India Census 2011 data, and from Shuttle Radar Topography Mission data version 4 respectively (see S3 File). Note that area of forest loss did not improve the overall accuracy of models. The administrative boundary dataset is from HindudstanTimesLabs (https://github.com/HindustanTimesLabs/shapefiles/), reproduced under the MIT License. Human case data are from Department of Health and Family Welfare Services, Government of Karnataka. White areas in Bhadravathi taluka in southeast corner indicate no data.

## Validating predictive models in an outbreak situation

The Boyce Index values i.e. the correlation between the predicted to expected ratio and the predicted probability of presence were uniformly high (r > 0.8) when all case locations in Shivamogga district or only those in the long-affected Tirthahalli taluka were considered in validation. For these areas, the Boyce Index was slightly higher for 1 km models than 2 km models and when area of forest loss was excluded from models (Tables S8A & S8B in S8 File)

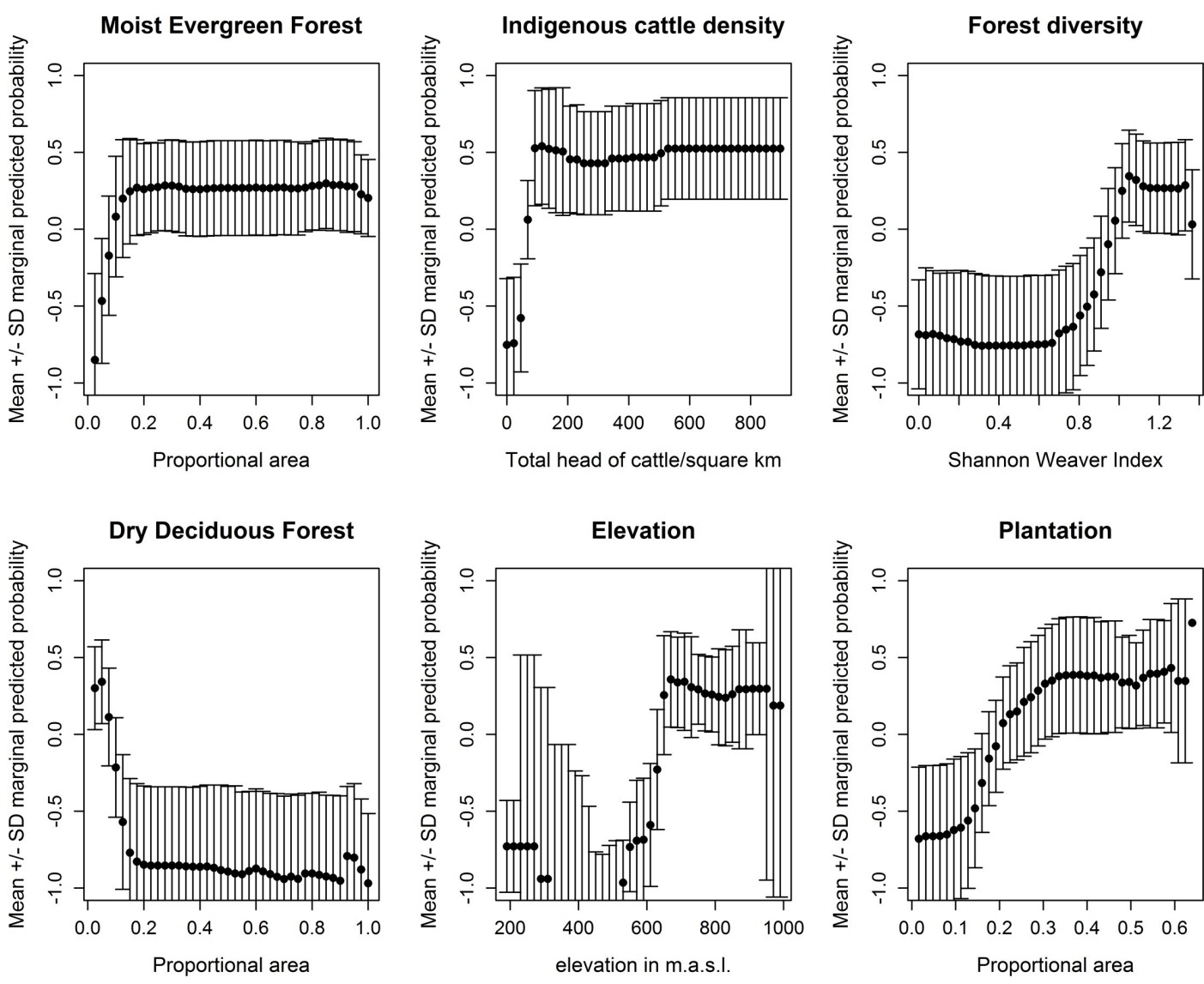

**Fig 4. Marginal response plots for key predictors of presence of human cases of Kyasanur Forest Disease, from models at a 1 km resolution (without forest loss as a predictor).**

suggesting that the predictions from these models are most consistent with the distributions of the presences in the independent validation datasets. This result was mirrored in the graphs of predicted versus expected ratios (S8 File) which were closest to a monotonic increase for the 1 km models without forest loss. When the model predictions for Tirthahalli are overlaid with the locations of cases from the 2018–2019 season (red circles, Fig 6), it can be seen that all but 2 of around 40 cases occur in yellow to red areas of medium to high predicted probability of presence.

Case locations in the newly affected Sagara taluka were less well captured by the models than case locations in the long-affected Tirthahalli taluka. Boyce Index values exceeded 0.8 for most model sets but were consistently lower for the Sagara data than the Tirthahalli data, (S8 File) and the graphs of predicted versus expected ratios were further from a monotonic increase (S8 File). The 2 km models excluding forest loss were most consistent with the

**Table 6. Relative importance of predictors in BRT models at a 2 km resolution (median, mean and s.d. of values across 100 model runs).**

| Predictor | 2 km resolution without forest loss metrics | | | | 2 km resolution with forest loss metrics | | | |
|---|---|---|---|---|---|---|---|---|
| | median RI | mean RI | s.d. RI | rank | median RI | mean RI | s.d. RI | rank |
| Area of moist evergreen forest | 23.8 | 24.5 | 12.7 | 1 | 20.5 | 21.5 | 12.6 | 1 |
| Area of forest loss | | | | | 11.9 | 14.5 | 9.4 | 2 |
| Area of dry deciduous forest | 13.8 | 16.2 | 9.3 | 2 | 11.8 | 14.6 | 8.8 | 2 |
| Area of plantation | 11.0 | 13.4 | 8.7 | 2 | 7.0 | 9.6 | 7.5 | 2 |
| Forest diversity | 8.3 | 9.6 | 6.7 | 5 | 6.9 | 8.0 | 6.0 | 6 |
| Elevation | 7.4 | 8.9 | 7.3 | 3 | 6.1 | 7.9 | 6.6 | 5 |
| Indigenous cattle density | 5.8 | 7.3 | 5.9 | 4 | 4.7 | 6.3 | 5.7 | 3 |
| Area of agricultural/fallow land | 4.0 | 4.9 | 3.5 | 6 | 3.4 | 4.1 | 3.0 | 6 |
| Area of moist deciduous | 3.8 | 5.3 | 4.5 | 8 | 3.4 | 4.8 | 4.4 | 9 |
| Slope | 2.6 | 3.0 | 2.1 | 9 | 1.9 | 2.4 | 1.8 | 10 |
| Area of waterbodies | 2.2 | 2.9 | 2.2 | 11 | 1.9 | 2.6 | 2.0 | 10 |
| Human population density | 1.5 | 1.9 | 1.6 | 11 | 1.3 | 1.7 | 1.5 | 13 |
| Buffalo | 1.1 | 1.6 | 1.5 | 12 | 1.1 | 1.5 | 1.6 | 13 |
| Proximity to public health centre | 0.2 | 0.5 | 0.7 | 13 | 0.2 | 0.4 | 0.5 | 14 |
| Number of medics per head of population | 0.0 | 0.1 | 0.3 | 14 | 0.0 | 0.1 | 0.3 | 15 |

distributions of the presences in the Sagara data. Though the major foci of human cases in central Sagara were captured by model predictions, several isolated case locations in the north, east and west occurred in grid cells predicted to have a low probability of presence of KFD (Fig 7).

Forest loss locations from the global dataset [46] did not correspond well with Landsat-derived estimates of current land use for Shivamogga (S3 File). Models including the area of forest loss per grid cell as a predictor had equivalent performance to models excluding area of forest loss in cross-validation, and generally performed worse in validation with the independent outbreak data (S8 File).

## Discussion

This study advances understanding of the landscape determinants of human cases of a zoonotic disease by considering wide-ranging topographical, host, land use factors and public health constraints. Human cases of Kyasanur Forest Disease tend to be highly localized, with epidemics lasting three to five years in a particular 30-40km$^2$ area, in which the disease shifts each season to affect a new handful of villages[28]. As seen in other tick-borne diseases[57], such highly focal patterns in human epidemics results from the interplay of environmental and ecological dynamics underpinning hazard and the human activities that govern exposure across the landscape [14,19].

Our models indicated that focal patterns in human KFD cases can be predicted with a high degree of accuracy from combined metrics of area and diversity of different forest types, plantation and cattle, as well as elevation. Overall, landscapes that were at high risk of occurrence of human KFD cases were characterised by diverse forest-plantation mosaics, containing large amounts of moist evergreen forest and plantation and low amounts of dry deciduous forests, with high indigenous cattle densities, occurring at elevations over 650 m a.s.l. These findings are consistent with prior suggestions that KFD is an "ecotonal" disease[8] and that creation of such habitat mosaics, when forest is removed for paddy cultivation and plantations, precipitates emergence of KFD in humans [25,26]. The models presented here quantify and map "risky" habitat explicitly.

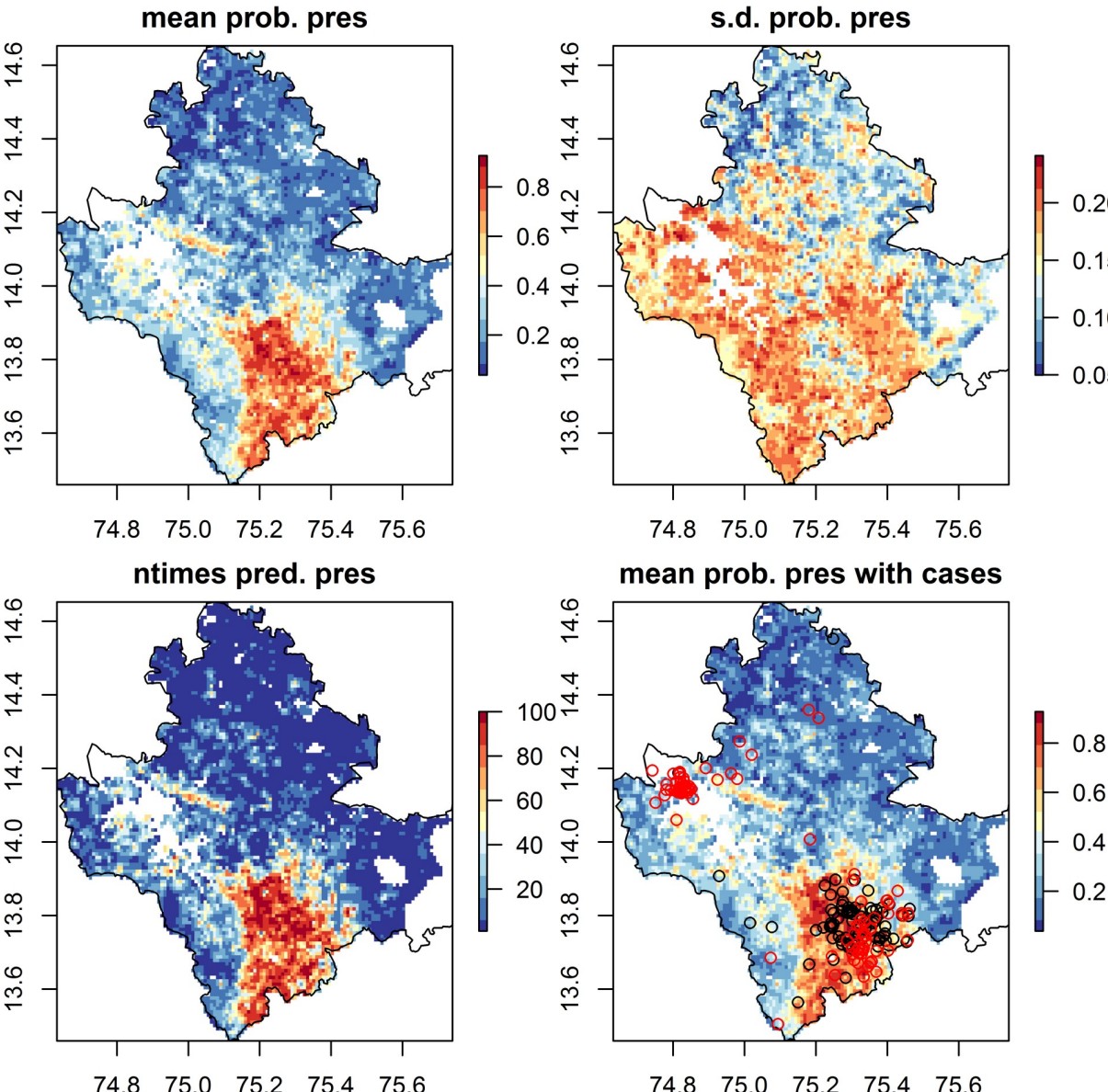

**Fig 5. Predicted probability of presence of KFD from Boosted Regression Tree models containing landscape predictors (without forest loss) at a 1 km resolution.** Top panels give mean and standard deviation of the relative predicted probability of presence of KFD in each grid cell across model runs. Top left panel—areas in orange and red have a higher predicted probability of presence of KFD, whilst areas in blue have a lower predicted probability of presence of KFD. Top right panel–areas in yellow and red have more variable predictions of probability of presence of KFD than blue areas. Bottom left panel indicates the number of times KFD is predicted to be present in each cell across the 20 model runs, with orange and red indicating that KFD is often predicted to be present. Bottom right panel again depicts the mean predicted probability of presence but the locations of affected households are super-imposed, for the 2014 to 2018 seasons as open black circles and for the 2018 to 2019 season as open red circles. These raster maps are not under copyright since they are a product of this study. The administrative boundary dataset is from HindudstanTimesLabs (https://github.com/HindustanTimesLabs/shapefiles/), reproduced under the MIT License. Human case data are from Department of Health and Family Welfare Services, Government of Karnataka. White areas in Bhadravathi taluka in south east corner indicate no data.

The models predicted human case locations with high accuracy in the sub-district from which the input data from the prior four years had been drawn. They also predicted large clusters of human cases in the newly affected sub-district over 30 km away. Their failure to predict some isolated case locations in the newly affected area, suggests that the models, parameterised

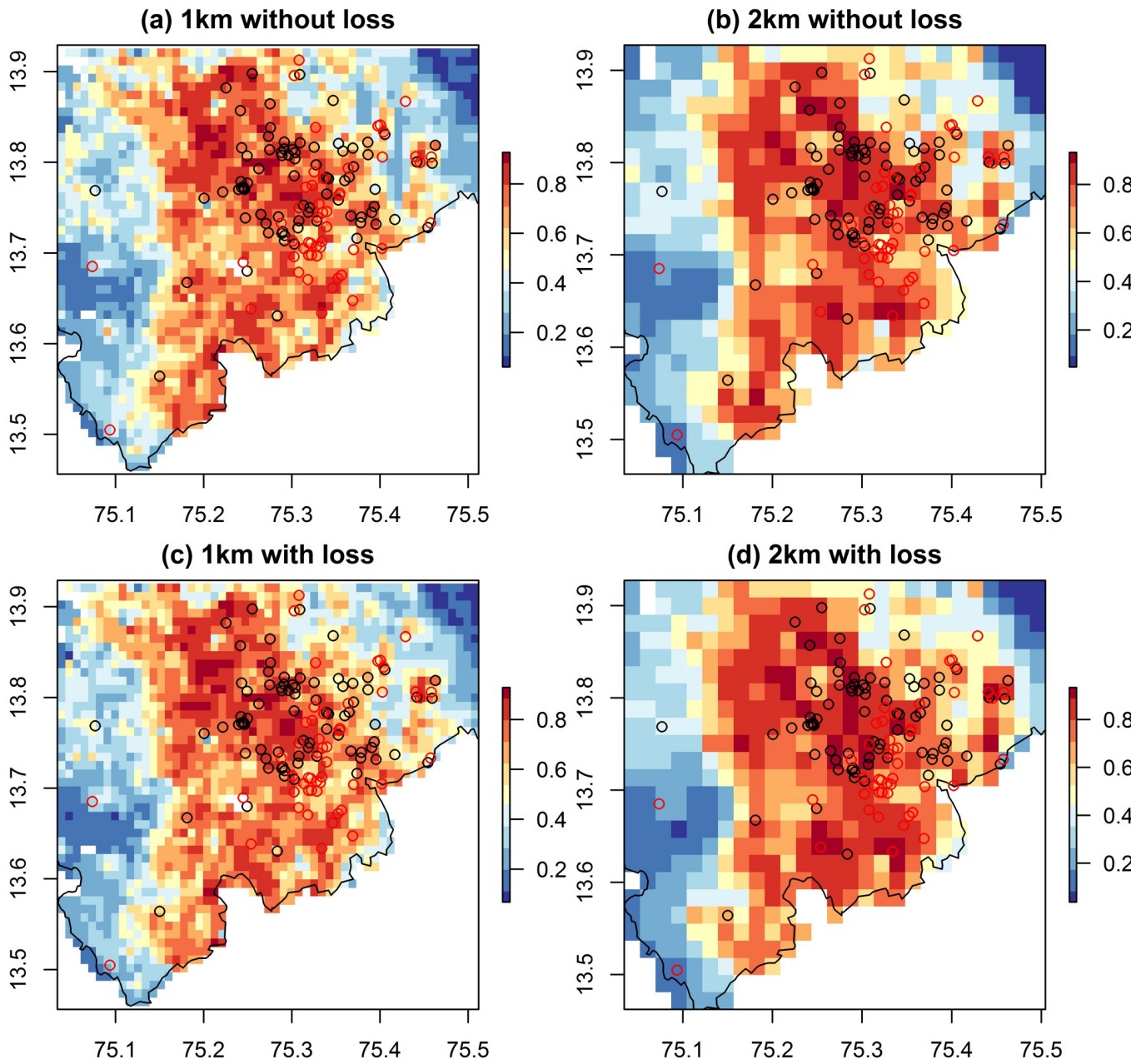

**Fig 6. Predicted probability of presence of KFD from Boosted Regression Tree models for the 2019 outbreak region in Tirthahalli taluk.** Models are at a 1 km (left hand panel) and 2 km resolution (right hand panel), with area of forest loss included in the predictors (bottom panel) and without (top panel). The outbreak locations from the 2018 to 2019 season (open red circles) occur mostly in areas of medium to high probability of presence (yellow to red) in all models, apart from two outbreak locations in the west. The black circles indicate outbreak locations from the 2014 to 2018 seasons, used to parameterise the models. White areas have missing environmental data because they contain water-bodies or are outside Shivamogga district (black outline). These raster maps are not under copyright since they are a product of this study. The administrative boundary dataset is from HindudstanTimesLabs (https://github.com/HindustanTimesLabs/shapefiles/), reproduced under the MIT License. Human case data are from Department of Health and Family Welfare Services, Government of Karnataka.

with four years data from an epidemic in one sub-district, may be under-estimating the "environmental niche" of human spill-over of KFD cases to some degree. Ideally, such correlative, pattern-matching models should be updated annually, preferably prior to the transmission season, as cases arise in new areas.

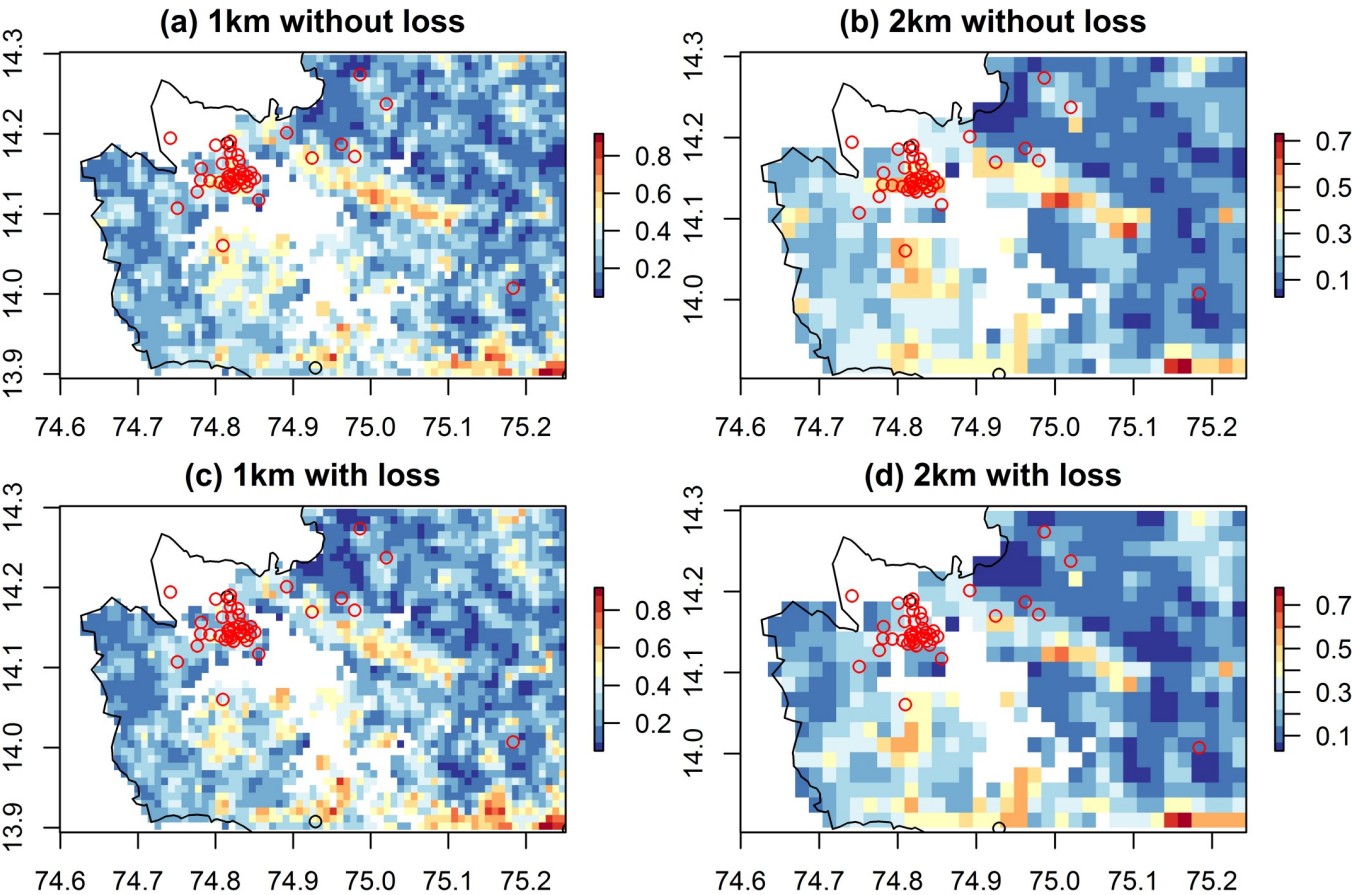

**Fig 7. Predicted probability of presence of KFD from Boosted Regression Tree models for the 2019 outbreak region in Sagara taluk.** Models are at a 1 km (left hand panel) and 2 km resolution (right hand panel), with area of forest loss included in the predictors (bottom panel) and without (top panel). The outbreak locations from the 2018 to 2019 season (open red circles), occur in areas of medium to high probability of presence (yellow to red) in all models. White areas have missing environmental data because they contain water-bodies or are outside of Shivamogga district. These raster maps are not under copyright since they are a product of this study. The administrative boundary dataset is from HindudstanTimesLabs (https://github.com/HindustanTimesLabs/shapefiles/), reproduced under the MIT License. Human case data are from Department of Health and Family Welfare Services, Government of Karnataka.

Currently, KFD management activities of vaccination, surveillance and awareness raising are conducted within a 5 km to 10 km radius of prior known cases (from the past 4 years). Given that vaccine resources are often constrained during the outbreak season and the uncertainty in our model predictions (depicted in Fig 5), we would not recommend that Health Departments rely on the resulting maps for spatial targeting of vaccine doses. Instead, pre-season tick surveillance activities of the Health Department and awareness raising in Primary Health Centres (early diagnosis of KFD) could be targeted towards areas of high relative predicted suitability of KFD that fall outside the 5 km radius of known cases. Thus would increase the likelihood that new foci of transmission are detected rapidly before or very early in the season, triggering rapid responses including vaccination.

Since model accuracy did not increase substantially when the area of forest loss was added to the models, this may indicate that the composition of forest and plantation types in the diverse mosaic around a settlement is of greater importance in determining KFD risk than a particular level of deforestation per se. This concurs with our a priori expectations that human contact rates across landscapes of different composition will depend on how human activities and the dynamics and infection rates of key vectors, wildlife and domestic reservoirs are linked

to different types of forest and cultivation. However, accurate quantification of the level of forest loss that can lead to emergence of KFD in humans was hampered by the poor correspondence between global spatial data on forest loss and gain [44] and the local patterns in forest types shown on the Land Use Land Cover map. The strong correlation between areas of forest gain and areas of forest loss in the global dataset for our study area suggests that the forest loss metric is more of an indicator of 'intensification', where forest is converted into plantation, rather than permanently lost (to make way for crops or built up areas). Local-scale understanding of these loss and intensification processes can be improved through time series analysis of Landsat Imagery (currently underway in the MonkeyFeverRisk project).

The finding that human cases of KFD are more likely to occur in landscapes with greater coverage of plantations is consistent with suggestions that large KFD epidemics in the 1970s and 1980s were linked to international development projects that replaced evergreen forest with cashew plantations [26]. It is also consistent with observations that migrant agricultural labourers in plantations have been widely affected in recent epidemics in Maharashtra and Goa States [27,58]. There are numerous potential explanations for the finding that human cases or KFD are more likely in landscapes with higher cattle densities. Cattle are thought to amplify local densities of ticks and bring ticks into households (either directly or through dry leaves used as fodder) whilst cattle grazing may bring small-holders into contact with tick habitat in forests, paddies and plantation.

Since our models are developed with disease data at household level, and scaled to the daily movement of local communities of 1–2 km through the landscape, impacts of micro-scale factors such as small water-bodies (<10 km) on disease patterns may not be detectable. Obtaining finer scale data on how locations of exposure to relate to such micro-scale factors is difficult because people are often unable to pinpoint where on their daily routes infected ticks were picked up and close monitoring of burdens by researchers may hamper livelihood activities.

The socio-ecological mechanisms linking human cases of KFD and landscape factors can be understood only through joint study across fragmented forest landscapes of the intersecting (1) human activities and priorities of forest users that underpin exposure and (2) vector and host population dynamics and infection rates that underpin hazard. Such empirical, inter-disciplinary, landscape scale studies for KFD are currently underway within the MonkeyFeverRisk project.

Prior studies have already highlighted several advantages of combining participatory methods and traditional models for understanding and predicting zoonotic diseases [59,60]. These include an improved understanding of contextual risk factors (including social, cultural, political and economic dimensions), both by corroborating a priori knowledge and highlighting hitherto unknown factors, and allowing models to be parameterised with realistic data at field level rather than with aggregated data (that is more often available from Public and Animal Health Systems). Here we highlight the (additional) benefits from linking participatory methods and models within a co-production process. Firstly, throughout the framing and knowledge integration stages, cross-sectoral stakeholders contribute valuable hypotheses about key risk factors and the policy landscape for zoonotic diseases that may have been over-looked in the scientific literature and can be tested within models (Table 1). For example, a key next step is to integrate remote sensing of seasonal dynamics of rice paddies into the MonkeyFeverRisk predictive model framework and field surveys since local disease managers highlighted these landscape features (and harvest time) as key potential nodes of interaction between people, wildlife hosts and tick vectors for KFD. Several risk factors highlighted by stakeholders, including poor data management, low vaccine uptake and under-reporting of monkey deaths, could not be integrated into models, either because they are not measured or because they are poorly linked to available geographical proxies. Awareness of these risk factors still informs treatment

of input data, interpretation of model outputs and priorities and partnerships for future collection of both health and environmental data. For example, spatial information on vaccine coverage is now being collected by Primary Health Centres and shared with District Health Officials whilst the Indian Government National Biodiversity Mission aims to address some gaps in data on alternative hosts and vectors for priority zoonotic diseases.

A second advantage of co-production in this context is that the spatial grain of models for zoonotic diseases was tailored to the scale at which forests are used, and to the size and distance between settlements, that govern how forest communities access health services and report cases of KFD. Thus co-production helped to bridge the traditional gap between the scale of model outputs and the scale of intervention and ecosystem use [17] and enabled finer scale relationships to be established between the environment and human disease cases (household-level). Many predictive models in the scientific literature that map geographical patterns in infectious diseases draw data from online databases such as ProMed and HealthMap which can often record only the locations of hospital or Primary health centres where cases are reported [40]. By working with disease managers to collate and interpret case data, it becomes clear that visits to health centres can occur far away from the settlements or forest habitat in which infection is acquired (e.g. for KFD may range from 5–10 km for small-holders but some 10s of km for tourists, students travelling between home and college, pilgrims and migrant labourers). If hospital or health centre data are used indiscriminately to parameterise disease-environment relationships, the resulting associations and predictive maps could be spurious and unsuitable for spatial targeting of interventions. Though these data drawbacks have long been noted for tick-borne diseases[61], probably due to the time required to collate fine scale data and engage with local disease managers, predictive maps are still being developed based solely on health centre locations and advocated as disease management tools, including for KFD [62]. We concur with Boden and McKendrick [63], who argued that all models supplied to health policy makers should adhere strictly to the principles of independence, transparency (autonomy), beneficence and justice. It is incumbent on disease modellers to appraise and be transparent about the suitability of available case data for relating and predicting infection processes from environmental conditions. The iterative, reflexive engagement with stakeholders' needs and knowledge, through a co-production process, can facilitate this transparency.

Finally, working directly with disease managers to interpret and validate model outputs means that the scale, appearance and explanation of resulting maps and guidance can be better tailored to their needs, increasingly the likelihood of uptake for spatial targeting of interventions. For example, cross-sectoral stakeholders in the MonkeyFeverRisk project indicated that predictive maps at scales from village level up to clusters of villages, with contextual landscape features like roads and household locations, would be most helpful for planning of vaccination, surveillance and awareness campaigns [37]. As a consequence, preliminary risk layers that could be visualised within Google Earth were supplied to the Shivamogga Health Department for validation and experimentation.

Our approach of using co-production to guide production of risk maps that integrate hazard and exposure factors influencing human disease, harnessing a broad range of stakeholder knowledge and expertise across sectors, represents an important step forward in managing zoonotic disease in LMICs. This approach is applicable across wide ranging individual and interacting zoonotic diseases affecting dependent communities in different ecosystems. It will be imperative to develop context-dependent co-production processes that account for the cultural and political dimensions that affect exposure through ecosystem use, alongside local environmental and ecological factors that determine hazard, and underpin success of inter-sectoral One Health collaboration.

## Supporting information

**S1 Fig. The distribution of pairwise distances between records of human cases of Kyasanur Forest Disease (2014–2018).**
(DOCX)

**S1 File. Scale of daily movements of people from their households into the forest for livelihood activities in a KFD-affected district.**
(DOCX)

**S2 File. Participatory methods to identify with cross-sectoral stakeholders the key risk factors for Kyasanur Forest Disease and key policies affecting transmission and management.**
(DOCX)

**S3 File. Environmental predictors of Kyasanur Forest Disease distribution.**
(DOCX)

**S4 File. Deriving the Land Use Land Cover map of Shivamogga from Earth observation data.**
(DOCX)

**S5 File. Marginal response plots for key predictors of presence of human cases of Kyasanur Forest Disease.**
(DOCX)

**S6 File. Test for spatial autocorrelation in residuals of Boosted Regression Tree models of Kyasanur Forest Disease.**
(DOCX)

**S7 File. Maps of predicted probability of presence of KFD from different sets of BRT models.**
(DOCX)

**S8 File. External validation of the risk maps using 2018/2019 outbreak data and the Boyce Index.**
(DOCX)

## Acknowledgments

DHFWS thanks the National Institute of Virology, Pune and the Manipal Centre for Virus Research for supporting them in the diagnostics of KFD for outbreak management.

## Author Contributions

**Conceptualization:** Bethan V. Purse, Narayanaswamy Darshan, Gudadappa S. Kasabi, Shivani K. Kiran.

**Data curation:** Narayanaswamy Darshan, Gudadappa S. Kasabi, France Gerard, Abhishek Samrat, Charles George, Abi T. Vanak, Meera Oommen, Mujeeb Rahman, Vijay K. Sandhya, M Mudassar Chanda, Shivani K. Kiran.

**Formal analysis:** Bethan V. Purse, Narayanaswamy Darshan, France Gerard, Abhishek Samrat, Charles George, Abi T. Vanak, Peter A. Henrys.

**Funding acquisition:** Bethan V. Purse, Gudadappa S. Kasabi, France Gerard, Abi T. Vanak, Meera Oommen, Sarah J. Burthe, Juliette C. Young, Prashanth N. Srinivas, M Mudassar Chanda, Manoj V. Murhekar, Subhash L. Hoti.

**Methodology:** Bethan V. Purse, Abi T. Vanak, Meera Oommen, Mujeeb Rahman, Sarah J. Burthe, Juliette C. Young, Prashanth N. Srinivas, Stefanie M. Schäfer, Vijay K. Sandhya, M Mudassar Chanda, Manoj V. Murhekar, Subhash L. Hoti, Shivani K. Kiran.

**Validation:** Bethan V. Purse, Narayanaswamy Darshan, France Gerard, Abhishek Samrat, Abi T. Vanak, Peter A. Henrys, Manoj V. Murhekar, Shivani K. Kiran.

**Visualization:** Bethan V. Purse.

**Writing – original draft:** Bethan V. Purse, France Gerard, Abhishek Samrat.

**Writing – review & editing:** Bethan V. Purse, Narayanaswamy Darshan, Gudadappa S. Kasabi, France Gerard, Abhishek Samrat, Charles George, Abi T. Vanak, Meera Oommen, Mujeeb Rahman, Sarah J. Burthe, Juliette C. Young, Prashanth N. Srinivas, Stefanie M. Schäfer, Peter A. Henrys, Vijay K. Sandhya, M Mudassar Chanda, Manoj V. Murhekar, Subhash L. Hoti, Shivani K. Kiran.

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
