## [Decision Letter · Decision Letter 0]

30 Oct 2019

Dear Dr. Purse:

Thank you very much for submitting your manuscript "Predicting disease risk areas through co-production of spatial models: the example of Kyasanur Forest Disease in India’s forest landscapes" (#PNTD-D-19-01239) for review by PLOS Neglected Tropical Diseases. Your manuscript was fully evaluated at the editorial level and by independent peer reviewers. The reviewers appreciated the attention to an important problem, but raised some substantial concerns about the manuscript as it currently stands. These issues must be addressed before we would be willing to consider a revised version of your study. We cannot, of course, promise publication at that time.

We therefore ask you to modify the manuscript according to the review recommendations before we can consider your manuscript for acceptance. Your revisions should address the specific points made by each reviewer. 

When you are ready to resubmit, please be prepared to upload the following:

(1) A letter containing a detailed list of your responses to the review comments and a description of the changes you have made in the manuscript.

(2) Two versions of the manuscript: one with either highlights or tracked changes denoting where the text has been changed (uploaded as a "Revised Article with Changes Highlighted" file); the other a clean version (uploaded as the article file).

(3) If available, a striking still image (a new image if one is available or an existing one from within your manuscript). If your manuscript is accepted for publication, this image may be featured on our website. Images should ideally be high resolution, eye-catching, single panel images; where one is available, please use 'add file' at the time of resubmission and select 'striking image' as the file type. 

Please provide a short caption, including credits, uploaded as a separate "Other" file. If your image is from someone other than yourself, please ensure that the artist has read and agreed to the terms and conditions of the Creative Commons Attribution License at http://journals.plos.org/plosntds/s/content-license (NOTE: we cannot publish copyrighted images). 

(4) If applicable, we encourage you to add a list of accession numbers/ID numbers for genes and proteins mentioned in the text (these should be listed as a paragraph at the end of the manuscript). You can supply accession numbers for any database, so long as the database is publicly accessible and stable. Examples include LocusLink and SwissProt.

(5) To enhance the reproducibility of your results, we recommend that you deposit your laboratory protocols in protocols.io, where a protocol can be assigned its own identifier (DOI) such that it can be cited independently in the future. For instructions see http://journals.plos.org/plosntds/s/submission-guidelines#loc-methods

While revising your submission, please upload your figure files to the Preflight Analysis and Conversion Engine (PACE) digital diagnostic tool, https://pacev2.apexcovantage.com/ PACE helps ensure that figures meet PLOS requirements. To use PACE, you must first register as a user. Then, login and navigate to the UPLOAD tab, where you will find detailed instructions on how to use the tool. If you encounter any issues or have any questions when using PACE, please email us at figures@plos.org.

We hope to receive your revised manuscript by Dec 29 2019 11:59PM. If you anticipate any delay in its return, we ask that you let us know the expected resubmission date by replying to this email.

To submit a revision, go to https://www.editorialmanager.com/pntd/ and log in as an Author. You will see a menu item call Submission Needing Revision. You will find your submission record there. 

Sincerely,

Emily Gurley

Deputy Editor

Andrew Azman

Deputy Editor

Reviewer's Responses to Questions

**Key Review Criteria Required for Acceptance?**

**Methods**

-Are the objectives of the study clearly articulated with a clear testable hypothesis stated?

-Is the study design appropriate to address the stated objectives?

-Is the population clearly described and appropriate for the hypothesis being tested?

-Is the sample size sufficient to ensure adequate power to address the hypothesis being tested?

-Were correct statistical analysis used to support conclusions?

-Are there concerns about ethical or regulatory requirements being met?

Reviewer #1: Objectives are clear and the study design appears appropriate to address the stated objectives. The data used are clearly described and the description of modelling analyses are well written.

Reviewer #2: Authors have analyzed the geo-spatial risk factors of an important zoonotic disease using a co-production approach. It has a potential to be of predictive value for focusing public health resources of surveillance for other zoonoses also. 

Methods used are mostly appropriate.

In the methods section the use of software such as 'R software' and other geo-spatial software should be explicitly mentioned in a separate paragraph.

lines 215-220 and 237-248 (human KFD case data):

The exact method used for geo-referencing for the cases of 2018-19 should be mentioned. Whether this involved GIS based geo-coordinate data for each patient or assigning locations based on google-earth/google-map platform after making a filed visit should be specified. The method has been outlined for earlier transmission season. It may be noted that the GIS based method during the field visit (using GIS enabled instruments or even smartphone) is more accurate.

The reason for not using a more accurate GIS-based geo-location strategy for the cases (especially for the recent transmission season) should be mentioned. Even for the past cases, GIS based geo-coordinates may be obtained by conducting household visits with the help of the frontline health workers.

The impact of accuracy of this key input data on the overall analysis and conclusions should be mentioned in the limitations section (which needs to be added by the authors). This is especially important since the authors themselves discuss the importance of micro-scale level features such as waterbodies in the range of 10 sq metre and the limitations of Landsat image at 30 m scale (lines 304-308)'

Reviewer #3: This manuscript describes the results of a study of risk factors for Kyasanur Forest Disease in the western Ghats of India. The authors identified potential risk factors to assess in models by involving stakeholders, who also participated in identifying the relevant scales for spatial analyses. The analyses identify several risk factors that are correlated with earlier outbreaks. The authors then identify future outbreak areas and compare these to an outbreak in 2018/19, concluding that the model accurately predicted the location of this outbreak.

This is an interesting study and the authors are investigating a neglected disease that is a major human health issue. The inclusion of stakeholders is a useful model for conducting an investigation of this type, and the partnership appears to have been a fruitful collaboration among investigators from different areas of research. 

My major concern with this study is that the authors do not take a critical enough look at their own data, at least based on the general presentation in the manuscript. For example, in several places, they note that the 2018/19 outbreak locations were entirely in zones of high risk, as predicted by their model. This is clearly not the case. Though some of the case locations are within high risk zones, others are in lower risk zones. Further, they indicate, e.g. in the abstract, that they correctly identified a hotspot of an outbreak before it occurred. However, in the region of Sagara taluk depicted in Figure 6, roughly 1/3 of the entire area is identified by the model as high risk, and the outbreak occurred in just one of the many locations they predicted as hotspots. It’s great news that there weren’t more outbreaks, but the authors do need to wrestle with the fact that their risk maps indicated a lot of hotspots, and approximately one of the many regions they identified actually had an outbreak. How should a resource-poor region deal with predictions like these? Would delivering vaccinations over such a large area strain resources? Might it be possible to refine their models with more/better data so that they could provide more accurate outbreak locations? Is this good enough to be useful? Is it good enough to avoid being counter-productive, e.g. by diffusing a public health response that could have been more focused with better data? Again, the larger concern here is that the authors risk coming across as advocating a particular approach rather than objectively evaluating its strengths and weaknesses. This is particularly problematic because their approach is not an unequivocal success. 

In another vein, the authors don’t address their inability to develop quantifiable indices for some of the variables identified as important by their stakeholders, e.g. Table 1, nor the fairly coarse fit between their spatial proxies for some of the variables and the identified risk factors. This is a major issue, and should be addressed in the Methods and in the Discussion, particularly in light of their discussion of the advantages of co-production. One of the challenges is that (appropriate) data might not be available. What issues does this raise? What kinds of partnerships could arise from such a discussion, i.e. to collect or gather some of the missing data? Would it be better to wait to run the models until those data can be obtained so that they more accurately reflect actual risk? 

I appreciate the clear explanation of both the need for and the response to spatial clustering of the case presence data, and the supplementary file that provides some additional information. Their conclusion is that there is still mild spatial autocorrelation, based on Moran’s I, but they conclude that it is not important despite its presence in fairly large percentages of their model runs. By what objective grounds do they determine that spatial autocorrelation is not a problem for their analysis? Also, it would be helpful for them to include the sample sizes of case sites after thinning.

The authors analyze their data with and without the incorporation of the variable for forest loss. They do not investigate the model with and without any other variables, nor do they rigorously evaluate the potential intercorrelation of their variables. These are important considerations and need to be addressed in the manuscript. In partcular, discussion topics like those on lines 621 and on, in which they interpret what they see as a lack of importance of forest loss as a predictor, are problematic. 

The authors mention in several places (e.g. line 570) in the manuscript that “When the model

predictions are overlaid with the outbreak locations in Sagarataluk in Fig. 6 (black locations),

it can be seen that all of these locations occur in orange to red areas of high predicted

probability of presence (except the south-eastern most isolated location which has a low

probability of presence in 2km models).” But this is clearly not the case in any of the maps presented. This undermines this reviewer’s confidence in the objectivity of their analysis.

**Results**

-Does the analysis presented match the analysis plan?

-Are the results clearly and completely presented?

-Are the figures (Tables, Images) of sufficient quality for clarity?

Reviewer #1: The analysis presented corresponds to the plan presented in the methods and results are clearly presented. In the general comments I make one suggestion with respect to showing the predictive maps without human case locations so that map underneath can be seen - suggest placing in a 4th plot.

Reviewer #2: Yes

Certain suggestions are as follows

Table 1 - row 8, column 4 - there is overlapping of text with the cell immediately below it. This needs to be corrected. Are the spatial proxies of 'low vaccination coverage' and 'poor diagnosis and surveillance' the same ?

Table 2 - the category of predictor can be clubbed and written once instead of repeating it

Table 4 - there is overlap between the last two columns. This needs to be correctly presented.

Reviewer #3: The manuscript would be improved by a thorough edit, e.g. line 86. 

Line 374 – Here, the methods indicate that the models are parameterised with presence-only data, though previously absence data were described. Please make a clearer distinction between when presence-only and presence/absence data were used to develop models. 

Line 275 – “Forest diversity” is not the ideal descriptor for the variable used here. Based on the description, they seem to have used “Diversity of forest types”, while “forest diversity” sounds like they quantified the species diversity within forests, which they didn’t. I would suggest changing this throughout. 

Line 621 – The authors discuss why incorporating forest loss did not substantially improve model accuracy. Their explanations do not incorporate a reason for why forest loss emerged as the second most important factor when it was incorporated. One possibility is that forest loss is correlated with another one of their variables, and I do not see where they rule out this possibility. 

Line 711 – Please spell out acronyms.

Figure 2. Why not replace the cryptic panel headings (e.g. Area_ME) that require a special sentence in the legend with informative headings right in the figure? (e.g. “Area of moist evergreen forest”).

Figure 6, line 981 – The authors claim that “The black outbreak locations occur in areas of high probability of presence (orange to red) in all models.” This statement is not actually consistent with the data presented in Figure 6, in which quite a number of the black dots occur in blue or white cells. 

For all figures with scales, more information about the scale values needs to be included.

**Conclusions**

-Are the conclusions supported by the data presented?

-Are the limitations of analysis clearly described?

-Do the authors discuss how these data can be helpful to advance our understanding of the topic under study?

-Is public health relevance addressed?

Reviewer #1: The discussion is well written, public health relevance is addressed and there is discussion on how the analyses presented can advance understanding of KFD distribution in addition to wider application of the methods used.

Reviewer #2: Conclusions (as mentioned in the discussion section) are supported by the data presented.

However, a brief conclusion section should be added after the discussion.

The limitations of the study have not been described and need to be discussed.

The relevance of the study has been discussed.

Reviewer #3: Overall, this is a potentially useful approach to modeling risk factors for an important disease, and the manuscript has many strengths. The co-production approach clearly had benefits, although it also exposed weaknesses of this approach which should be addressed, as should the weaknesses of their predictive abilities based on model outputs.

**Editorial and Data Presentation Modifications?**

Reviewer #1: (No Response)

Reviewer #2: modifications suggested are as follows:

Longer sentences may be reframed and converted to shorter sentences for better comprehension and readability ( examples - lines 681-684 and 704-709)

All R function names such as 'correlog' may be italicized.

Space may be added between number and km throughout, i.e. '1km' may be written as '1 km'.

215 - it is not necessary to write the co-author within the article. Rather, the specific contribution can be elaborated within the section of 'Author contributions'

240 - edit duration as ' 11 December 2018 - 12 January 2019'

242, 571 and 575 - kindly add space after 'Sagara'

596 - 'hosts' should be replaced by 'host-related'

600 - 'inter-play' should be replaced by 'interplay'

605 - consider replacing sentence fragment as ' Overall, landscapes that were at high risk of emergence of human KFD cases...'

610- consider adding i.e. after mosaics,

631 - consider insertion of 'of' between 'more' and 'an'

657 - consider insertion of 'using' between 'than' and 'aggregated data'

683 - consider use of more technically valid adjectives such as 'insufficiently predictive' instead of 'poor'

701 - consider replacing comma by 'of'

704 - use abbreviation 'LMIC' instead (abbreviated already)

Reviewer #3: (No Response)

**Summary and General Comments**

Reviewer #1: This study aimed to develop spatially-explicit predictive models of Kyansur Forest disease (KFD) occurrence for an endemic region of India. The choice of risk factors for inclusion in models was based on information gained from a workshop with key stakeholders. Machine learning algorithms were used to fit models to human case data. Resulting models were then tested for the ability to identify areas where a new outbreak had occurred, not used for initial model fitting. The modelling identified diverse forest-plantation mosaics with high proportion of moist evergreen forest, high cattle density and elevation above 650m as important risk factors for KFD.

Overall, I think that the manuscript is of publishable quality and suitable for PLoS NTDs. In particular, the combined use of: i) information from key stakeholders to guide modelling and; ii) case data from a new area to validate models is novel with respect to zoonotic diseases relevant to developing countries. I do not have any major concerns with respect to the scientific approach and conclusions made but highlight that my skills and experience are not sufficient to comment in detail on the modelling techniques used.

The manuscript overall is well structured and well written. I only really therefore have three suggestions:

The manuscript could be shortened substantially, in particular the introduction. I would suggest cutting the text on lines 117 – 126 about ‘co-production’ and then lines 164 – 187 and instead replace with concise aims of the project which I interpret to be: i) identification of risk factors to include in models through stakeholder engagement; ii) development of predictive models; iii) validation of models. I do think the text/ framing around ‘co-production’ over complicates the paper and instead reference to ‘one-health’ approaches would suffice and simplify. 

In relation to the above, suggest changing the title to be explicit about what was done and what the study found. Perhaps something along the lines of ‘Spatial models developed through stakeholder engagement predict high risk areas of KFD in India’.

For figures showing predictive maps and human case locations perhaps have open circles or show cases separately on a 4th panel so that the map beneath can be seen clearly.

Reviewer #2: Article is well-written overall. However certain portions in the results section needs to be shortened to avoid repetition with data presented in figures and tables. Longer sentences may be may be reframed into shorter sentences.

Unnecessary use of brackets (such as 349, 382 ) may be avoided to maintain the flow of the sentence.

Reviewer #3: (No Response)

PLOS authors have the option to publish the peer review history of their article (what does this mean?). If published, this will include your full peer review and any attached files.

Reviewer #1: No

Reviewer #2: Yes: Dr Suman Saurabh

Reviewer #3: No

---

## [Decision Letter · Decision Letter 1]

21 Feb 2020

Dear Dr. Purse,

Thank you very much for submitting your manuscript "Predicting disease risk areas through co-production of spatial models: the example of Kyasanur Forest Disease in India’s forest landscapes" for consideration at PLOS Neglected Tropical Diseases. As with all papers reviewed by the journal, your manuscript was reviewed by members of the editorial board and by several independent reviewers. The reviewers appreciated the attention to an important topic. Based on the reviews, we are likely to accept this manuscript for publication, providing that you modify the manuscript according to the review recommendations, which are primarily editorial. 

Sincerely,

Emily Gurley

Deputy Editor

Andrew Azman

Deputy Editor

The authors have responded to reviewer comments and as a result, the manuscript is improved. However, there are a few editorial changes that reviewers suggest prior to accepting the manuscript.

Reviewer's Responses to Questions

**Key Review Criteria Required for Acceptance?**

**Methods**

-Are the objectives of the study clearly articulated with a clear testable hypothesis stated?

-Is the study design appropriate to address the stated objectives?

-Is the population clearly described and appropriate for the hypothesis being tested?

-Is the sample size sufficient to ensure adequate power to address the hypothesis being tested?

-Were correct statistical analysis used to support conclusions?

-Are there concerns about ethical or regulatory requirements being met?

Reviewer #1: (No Response)

Reviewer #2: Objectives are clearly articulated and methods are appropriate

**Results**

-Does the analysis presented match the analysis plan?

-Are the results clearly and completely presented?

-Are the figures (Tables, Images) of sufficient quality for clarity?

Reviewer #1: (No Response)

Reviewer #2: Results are clearly presented

**Conclusions**

-Are the conclusions supported by the data presented?

-Are the limitations of analysis clearly described?

-Do the authors discuss how these data can be helpful to advance our understanding of the topic under study?

-Is public health relevance addressed?

Reviewer #1: (No Response)

Reviewer #2: Conclusions are supported by the data and limitations have been discussed.

**Editorial and Data Presentation Modifications?**

Reviewer #1: Check for consistency in unit placement after values e.g. line 249: 2km Line 223: 330 m

Line 265: remove full stop

Table 1: all column contents start with capital letter except ‘proximity to health centre…’

Table 2: again, consistency with use of capitals, would also have each text in column two align with top of text relevant to each row in column three

Line 382: remove comma after full stop.

Line 443: Should be Liverpool School of Tropical Medicine (no Hygiene)

Table 3: again consistency with use of capitals

Line 557: insert 'be' in ‘is often predicted to present’

Reviewer #2: Minor edits suggested-

Line 381 - 'adhoc' to be changed to 'ad hoc', with addition of intervening space

Line 382 - space to be added after 'selected'

Line 382 - Consider removing comma immediately after full-stop.

Line 385 -386 (choice of punctuation marks) - a '-' and ','can be used instead of colon and semi-colon respectively.

Line 421 - remove space between 'to-' and 'expected' , thereby making 'predicted-to-expected' in continuity.

Lines 517-521 - the frequent use of semi-colon (;) in the sentence needs to be reconsidered. For example, there appears to be no need of semi-colon after 'when' in line 517. Other semi-colon may be replaced by comma.

Lines 688-692 - inclusion of '(1)' and '(2)' within the sentence are not necessary.

Line 756-761 - The last sentence in discussion section appears to be an inordinately long sentence. Needs to be re-framed/ shortened. Similarly 688-692 and 676-680 are examples where the sentence used appears to be quite long. It appears to me that average sentence length is longer in this article than what would be needed for easy comprehension by readers.

Likewise, authors are suggested to review the manuscript for minor corrections and proper presentation of text.

**Summary and General Comments**

Reviewer #1: I accept the authors points about 'co-production' in repsonse to my initial comments. I do not have any further comments and think that this draft is suitable for publication.

Reviewer #2: The suggested changes have been incorporated. Where not incorporated, proper justification has been provided. Manuscript is much improved as a result. 

Authors have done well to have a comprehensive data analysis. Since the write-up is long, the possibility for minor errors and consequently, the scope for improvement is also present. However, a small additional effort in making the write-up crisp and free from minor errors will boost the quality of the article.

PLOS authors have the option to publish the peer review history of their article (what does this mean?). If published, this will include your full peer review and any attached files.

Reviewer #1: No

Reviewer #2: Yes: Dr Suman Saurabh
---

## [Editor Report · Decision Letter 2]

27 Feb 2020

Dear Dr. Purse,

We are pleased to inform you that your manuscript 'Predicting disease risk areas through co-production of spatial models: the example of Kyasanur Forest Disease in India’s forest landscapes' has been provisionally accepted for publication in PLOS Neglected Tropical Diseases.

Before your manuscript can be formally accepted you will need to complete some formatting changes, which you will receive in a follow up email. A member of our team will be in touch within two working days with a set of requests.

Best regards,

Emily Gurley

Deputy Editor

Andrew Azman

Deputy Editor

---

## [Editor Report · Acceptance letter]

1 Apr 2020

Dear Dr. Purse,

We are delighted to inform you that your manuscript, "Predicting disease risk areas through co-production of spatial models: the example of Kyasanur Forest Disease in India’s forest landscapes," has been formally accepted for publication in PLOS Neglected Tropical Diseases.

Best regards,

Serap Aksoy

Editor-in-Chief

Shaden Kamhawi

Editor-in-Chief
